# Genome-Scale Investigation of *GARP* Family Genes Reveals Their Pivotal Roles in Nutrient Stress Resistance in Allotetraploid Rapeseed

**DOI:** 10.3390/ijms232214484

**Published:** 2022-11-21

**Authors:** Ying-Peng Hua, Peng-Jia Wu, Tian-Yu Zhang, Hai-Li Song, Yi-Fan Zhang, Jun-Fan Chen, Cai-Peng Yue, Jin-Yong Huang, Tao Sun, Ting Zhou

**Affiliations:** 1School of Agricultural Sciences, Zhengzhou University, Zhengzhou 450001, China; 2School of Life Sciences, Zhengzhou University, Zhengzhou 450001, China

**Keywords:** *Brassica napus*, transcription factors, nutrient stress, transcriptomic analysis, miRNA

## Abstract

The *GARP* genes are plant-specific transcription factors (TFs) and play key roles in regulating plant development and abiotic stress resistance. However, few systematic analyses of *GARPs* have been reported in allotetraploid rapeseed (*Brassica napus* L.) yet. In the present study, a total of 146 *BnaGARP* members were identified from the rapeseed genome based on the sequence signature. The *BnaGARP* TFs were divided into five subfamilies: *ARR*, *GLK*, *NIGT1/HRS1/HHO*, *KAN*, and *PHL* subfamilies, and the members within the same subfamilies shared similar exon-intron structures and conserved motif configuration. Analyses of the Ka/Ks ratios indicated that the *GARP* family principally underwent purifying selection. Several *cis*-acting regulatory elements, essential for plant growth and diverse biotic and abiotic stresses, were identified in the promoter regions of *BnaGARPs*. Further, 29 putative miRNAs were identified to be targeting *BnaGARPs*. Differential expression of *BnaGARPs* under low nitrate, ammonium toxicity, limited phosphate, deficient boron, salt stress, and cadmium toxicity conditions indicated their potential involvement in diverse nutrient stress responses. Notably, *BnaA9.HHO1* and *BnaA1.HHO5* were simultaneously transcriptionally responsive to these nutrient stresses in both hoots and roots, which indicated that *BnaA9.HHO1* and *BnaA1.HHO5* might play a core role in regulating rapeseed resistance to nutrient stresses. Therefore, this study would enrich our understanding of molecular characteristics of the rapeseed *GARPs* and will provide valuable candidate genes for further in-depth study of the GARP-mediated nutrient stress resistance in rapeseed.

## 1. Introduction

The transcriptional regulation of plant genes is a complex and accurate network system. In this process, transcription factors (TFs) play crucial roles in plant growth and development, species origin, and stress responses by precisely binding to the *cis*-acting regions of target genes [1]. After the identification of the Arabidopsis genome, the TFs were classified into 58 TF families [2]. Plant responses to nutrient stresses are regulated by complex signaling pathways and networks which are coordinated by TFs [3].

The *GARP* gene is a plant-specific TF and plays a key role in regulating plant development, disease resistance, hormone signaling, circadian clock oscillations, and abiotic stress resistance [4]. *GARP* is named from the Golden 2 (G2) protein in *maize*, the type B authentic response regulator (ARR-B) protein in *A. thaliana*, and the phosphate starvation response 1 (*PSR1*) protein in *Chlamydomonas* [5]. In the *GARP* family, the members can be classified if the derived protein contains the conserved signature motif called the B-motif (GARP motif) [6]. The B motif is a signature of type-B response regulators (ARRs) involved in His-to-Asp phosphorelay signal transduction systems in Arabidopsis, which contains an HTH (helix-turn-helix) motif [5]. HTH motifs can regulate a variety of physiological processes, as well as participate in TF dimerization. The B motif is highly similar to MYB-DBD (Myb-DNA binding domain), and this also leads to frequent confusion with MYB-related TFs [5]. In contrast to MYB-related proteins characterized by the (SHAQK(Y/F) F) motif, GARP TFs contain a different consensus sequence (SHLQ(K/M) (Y/F)) [6].

The *GARP* TFs have been identified in Arabidopsis, rice, cotton, tea plant, and other species, and related studies have shown that they are involved in the regulation of plant growth and development, abiotic stress resistance, and other biological processes [7,8,9,10]. *AtGARPs* have been defined as important regulators of diverse nutrient stresses. The expression of *AtHHO3* (*NIGT1.1*) and *AtHHO2* (*NIGT1.2*) was induced in nitrogen (N) deficiency, while *AtHHO1(NIGT1.3*) is involved in primary root shortening under phosphate (Pi)-deficient conditions [11]. *AtKAN1* acts as a transcriptional repressor involved in auxin biosynthesis, auxin transport, and auxin response [12], and *AtKAN4* is shown to broadly control the flavonoid pathway in Arabidopsis seed [13]. In addition, *AtBOA* is a component of the Arabidopsis circadian clock [14]. In rice, *OsPHR1-4* has been linked to controlling Pi homeostasis-regulating sensing and signaling cascades in rice [15]. The expression of *OsARR-B5, OsARR-B22*, and *OsARR-B23* was upregulated under alkaline stress and was implicated in plant development modulation by controlling cellular processes, molecular activities, and biological functions [16]. *OsHHO2* inhibits Pi starvation response [17], whereas *OsHHO3* and *OsHHO4* play critical roles in the N deficiency response [18]. In cotton, *GhGLK1* is reported to be involved in the regulation of drought and cold stress responses [19]. Thus, GARP TFs play significant roles in the responses of different plant species to nutrient stress.

The GARP TFs are involved in the responses to nutrient stresses and include probable nutrient sensors of plants. CrPsr1 is the first reported GARP TF to be involved in the nutritional responses, and it is essential for the adaptation of *C. reinhardtii* to Pi starvation [20]. Then, NIGT1/HRS1/HHOs were found to be the most robustly and quickly nitrate (NO_3_^−^) regulated TFs [21]. N and Pi are essential macronutrients for the growth and development of plants. N participates in a variety of physiological and biochemical processes as a component of proteins, nucleic acids, and plant growth regulators [19]. Pi is an essential building block of Important compounds such as DNA, RNA, and proteins, and is involved in glycolysis, respiration, and photosynthesis [22]. In the process of N and Pi absorption and utilization, TFs play an important role in regulation [23,24,25,26]. A number of *GARPs*, particularly members of the *NIGT1/HRS1/HHO* subfamily, have been shown to play important roles in the regulation of plant responses to N and Pi stresses. The role of *NIGT1/HRS1/HHO* in response to N and Pi stresses can be summarized in two pathways: NRT-NLP-NIGT1/HRS1/HHO and NIGT1-SPX-PHR [27]. Specifically, the nitrate transporters *NRTs* can increase the content of N nutrients, enhancing the expression of the nitrate-responsive nodulin-like proteins (*NLPs*), and induce the expression of *GARPs* to suppress the N starvation response. The NIGT1 proteins repress the expression of *SPXs* by directly binding to the *SPX* promoters, and the SPX proteins function as the repressors of *PHR* TFs [28]. Under Pi-sufficient conditions, *PHR1* interacts with *SPXs* (SYG1/PHO81/XPR1), Pi sensor proteins, and inhibitors for *PHR1*, and the NIGT1-clade genes are not activated. Under Pi starvation conditions, *PHR1* is released from *SPXs* and promotes the expression of the NIGT1-clade genes [29]. In addition to *NIGT1/HRS1/HHO*, *PHR* of the *GARP* gene family has also been reported to affect N and Pi homeostasis [15].

Rapeseed (*Brassica napus* L.) is a major oilseed crop due to its economic value and oilseed production. However, its productivity has been repressed by many environmental adversities [30]. Under drought tolerance, the shoot and root growth of rapeseed seedlings is greatly inhibited, which ultimately will reduce crop production [31]. For rapeseed, salt stress severely affects all life stages from seed germination to yield production [32]. Cold stress has a negative impact on rapeseed germination and seedling establishment, causing wilting and plant death at the seedling stage [33]. Rapeseed production in the field is also often severely inhibited due to N deficiency [34] and is highly dependent on N fertilizer application, but its N use efficiency (NUE) is very low [35]. Rapeseed is also extremely sensitive to Pi deficiency [36]. A number of gene families, such as superoxide dismutase (*SOD*) [37], lipid phosphate phosphatases (*LPP*) [38], and B-box (*BBX*) [39] play critical roles in rapeseed growth, development, and response to stresses in rapeseed. To date, *GARPs* have also been identified to play important roles in plant growth and response to stress [16,17,18]. Given the importance of the *GARP* family in all aspects of plant developmental processes and stress responses, a comprehensive genome-wide investigation of *GARPs* is warranted in rapeseed.

However, few systematic analyses of *GARPs* in *B. napus* have been available so far. Thus, this study is aimed to (i) identify the genome-wide *GARPs* in *B. napus*, (ii) characterize the genomic characteristics and transcriptional responses of the *GARPs* to N stresses (including NO_3_^−^ limitation and ammonium (NH_4_^+^) toxicity) and Pi limitation, and (iii) investigate the transcriptional responses of *GARPs* to other nutrient stresses, including boron deficiency, cadmium toxicity, and salt stress. This study would enrich our understanding of molecular characteristics of the rapeseed *GARPs* and will provide valuable candidate genes for further in-depth study of the *GARP*-mediated nutrient stress resistance in rapeseed.

## 2. Results

### 2.1. Genome-Wide Identification of the GARP Family Genes in B. napus

Since the GARP proteins are highly similar to MYB or MYB-like TFs in terms of both sequences and structures, the candidate *GARP* genes were compared and screened according to the methods reported by Safi et al. [6]. In this study, a total of 146 *BnaGARPs* were identified in the rapeseed genome (A_n_A_n_C_n_C_n_: A1-A10, C1-C9).

The physical and chemical characteristics, including the gene length and molecular weights (MW), of a total of 146 GARP proteins were analyzed and provided. The *BnaGARPs* have varying physicochemical characteristics (Appendix A). The length of the GARP protein sequences ranged between 101 (*BnaA3.MYBC1a*) and 1022 (*BnaC8.GLK1*) amino acids in *B. napus*. The isoelectric point (pI) ranged between 4.74 (*BnaA2.PHL1*) and 11.06 (*BnaA5.MYBC1*). The molecular weight (MW) ranged from 11.62 to 111.69 kDa for *BnaA3.MYBC1a* and *BnaC8.GLK1*, respectively.

### 2.2. Phylogenetic Analysis and Ka/Ks Ratio Calculation

To elucidate the evolutionary relationships and functional divergence among Brassica GARP proteins, the sequences of 146 *B. napus* GARP proteins, and 56 *A. thaliana* GARP proteins were used to construct a phylogenetic tree (Figure 1). In general, the *GARP* homologs in *B. napus* significantly expanded compared to those in *A. thaliana*. Moreover, the number of *GARPs* in *B. napus* is much more than three times of those in *A. thaliana*.

Based on the topologies and bootstrap support values of the NJ phylogenetic tree, the candidate GARPs were divided into five subfamilies, which were identical to the previous study [7]. The distribution of *BnaGARPs* among different subfamilies was as follows: *ARR* (34 members), *GLK* (17 members), *NIGT1/HRS1/HHO* (27 members), *KAN* (17 members), and *PHL1* (52 members). The differences in the number of *BnaGARPs* within the five subfamilies indicated a distinct expansion trend among these subfamilies.

To explore the selective pressure on *BnaGARPs*, the non-synonymous/synonymous mutation ratio (Ka/Ks) was calculated; Ka/Ks > 1.0 indicates positive selection, Ka/Ks = 1.0 indicates neutral selection, and Ka/Ks < 1.0 indicates purifying selection [40]. The Ka/Ks ratio for all *BnaGARPs* was <1.0, ranging between 0.0697 (*BnaA5.ARR1*) and 0.5771 (*BnaA6.APRR4*), implying that the replicated *GARPs* could experience strong purification selection (Appendix A).

### 2.3. Conserved Motif and Gene Structure Analyses

To further clarify the potential functions of *GARPs* in *B. napus*, MEME was used to identify 10 conserved motifs (Figure 2). Motif 1 and motif 2 in the *BnaARR-B* subfamilies, and the motif1 and motif 4 in the *BnaNIGT1/HRS1/HHO* subfamilies are the B-motif of the *GARP* signature motif and extensively distributed in *BnaGARPs* (Figure 2B). Furthermore, motif 7 and motif 10 only exist in *APRR2*, while motif 8 is specific to *BnaARR2 and BnaARR1*, and in the PHL subfamily, motif 8 only occurs in *PHL14* (Appendix A). However, the motif patterns of *BnaGARPs* within a subgroup are similar.

To evaluate the sequence diversity of *BnaGARPs*, the exon–intron structures of each *BnaGARP* were detected. In detail, most of *BnaGARPs* had six exons and five introns, and several genes had five exons and four introns, while *BnaARR21s* contained 12 introns and *BnaMYBC1* contained one intron (Appendix A). Similarly, the majority of *BnaGARPs* in the same subgroups generally had similar gene structures (Figure 3).

We also found that the intron lengths are slightly different among different *BnaGARPs*. In comparison with *BnaPHL12s*, the introns within *BnaPHL5s* were relatively large. Although the exon-intron structures of most closely related genes exhibited high similarity and conservation, there still exist several differences.

### 2.4. Gene Duplication and Synteny Analysis of GARP Gene Families

The genomic positions of the identified *GARPs* were physically mapped onto the rapeseed chromosomes using the MapGene2Chrom program. Ultimately, a total of 146 *GARPs* were mapped onto 20 chromosomes in *B. napus* (Figure 4). Evidently, there are only two *GARPs* on chromosomes chrA4, four on chrA10, while chrC4 has the most genes. The distribution of genes on the chromosomes is relatively scattered, and the genes on the same chromosome are far apart.

Gene duplication events can lead to the expansion of gene families and play crucial roles in the adaptation of plant species to the external environment by acquiring new gene functions. Given the importance of gene duplication in the evolution of plant gene families, the duplication patterns of 146 *GARP* family genes were analyzed in *B. napus*. Between the two sub-genomes of *B. napus*, 45 duplication events took place on the A subgenome, 36 events on the C subgenome, and 136 events across the A/C subgenomes (Appendix A).

To better understand the evolution of *BnaGARPs*, the synteny of the *GARP* pairs between the genomes of *B. napus* and *A. thaliana, G. max, and M. truncatula* was constructed (Figure 5 and Appendix A). We found that 115 *BnaGARPs* exhibited syntenic relationships with *AtGARPs.* Some *AtGARPs* were associated with more than one orthologous copy in *B. napus.* For example, *KAN2/AT1G32240* showed a syntenic relationship with *BnaC5.KAN2b*, *BnaC5.KAN2a*, and *BnaC8.KAN2* (Appendix A). As shown in Figure 5 and Appendix A, *BnaGARPs* shared 172 syntenic gene pairs with *G. max,* 66 with *M. truncatula,* and 3 with *T. aestivum* (Appendix A). Additionally, syntenic gene pairs were identified between rapeseed and rice, which constituted the fewest number of background collinear blocks (Appendix A). Interestingly, 46 genes were found in the comparative synteny maps between *B. napus* and other plant species (*A. thaliana, M. truncatula, and G. max*), and these collinear gene pairs were highly conserved within several syntenic blocks, such as *BnaA1.APRR2, BnaA1.HHO5, BnaA1.KAN3, BnaA1.PHR1, and BnaA10.MYR1* on the A1 chromosome and *BnaA3.APRR2, BnaA3.ARR10,* and *BnaA3.ARR2* on the A3 chromosome.

### 2.5. Cis-Regulatory Element Prediction in the Promoter Regions of BnaGARPs

To investigate the potential regulatory mechanisms underlying *GARPs* in response to abiotic stresses and hormones, the *cis*-regulatory elements (CREs) in the 2000 bp upstream promoter sequences of each *GARP* gene were scanned by the plantCARE database. The results revealed that the promoter regions of each *BnaGARPs* have stress and hormone-related CREs.

In total, 20 types of CREs were detected, including 1848 light responsiveness CREs, 442 MeJA responsiveness CREs, 372 abscisic acid responsiveness CREs, and 365 anaerobic induction CREs (Appendix A). The most and least CREs found in the promoter regions of the *GARP* genes were light responsiveness CREs (1848) and wound-responsive CREs (3), respectively. Meanwhile, a mass of putative CREs that were involved in hormone responses, such as GA, MeJA, and BHA, were found in a series of *BnaGARP* promoters. As well, many putative CREs associated with abiotic stress, such as the low-temperature responsive CREs, defense and stress responsive CREs, and drought inducibility CREs, were found in many *BnaGARP* promoter regions (Figure 6).

### 2.6. Genome-Wide Analysis of miRNA Targeting BnaGARPs

In plants, miRNAs are important regulators of gene expression and play pivotal roles in abiotic stress responses [41]. To identify whether miRNAs are involved in the regulation of the *BnaGARP* expression, we identified 29 putative miRNAs targeting 34 *BnaGARPs* (Figure 7). Some of the miRNA-targeted sites are presented in Appendix A, while the detailed information of all miRNAs targeted genes is presented in Appendix A. The results showed that four members of the bna-miR164 family targeted three *BnaGARPs* (including *BnaC3.ARR1, BnaC6.HHO2*, and *BnaA7.HHO2*). Four members of the bna-miR172 family targeted two *BnaGARPs* (including *BnaC2.ARR18a* and *BnaC2.ARR18b*). Three members of the bna-miR390 family targeted two *BnaGARPs* (including *BnaA3.PHL2* and *BnaC3.PHL2*). Two members of the bna-miR397 family targeted three *BnaGARPs* (including *BnaC1.PHR1*, *BnaA1.PHR1*, and *BnaC7.PHR1*). Three members of the bna-miR156 family targeted *BnaC2.ARR18b.* One member of the bna-miR6029 family targeted four *BnaGARPs* (including *BnaA6.HRS1*, *BnaC5.HRS1*, *BnaA6.PHL6*, and *BnaCnn.PHL6*). One member of the bna-miR860 family targeted eight *BnaARRs* (Figure 7; Appendix A). Predominantly, *BnaA3.ARR2*, *BnaC3.ARR1*, *BnaC6.HHO2* and *BnaC2.ARR18b* were predicted to be targeted by several miRNAs (Figure 7; Appendix A).

### 2.7. Transcriptional Analysis of BnaGARPs under N and Pi Stresses

Nitrogen (N) is an essential macronutrient for plant growth and development, whereas rapeseed has a low NUE [42]. To improve the understanding of the role of *BnaGARPs* in NUE regulation in *B. napus*, the transcriptional responses of *BnaGARPs* were explored under low N conditions. Under limited NO_3_^−^ conditions, 40 members of *BnaGARPs* were differentially expressed in rapeseed plants compared to sufficient NO_3_^−^ (Figure 8). In the *BnaNIGT1/HRS1/HHOs* subfamily, most members were downregulated (87.88–98.12%) in the shoots under low NO_3_^−^ supply. Notably, the expression levels of *BnaC7.HHO3* and *BnaA9.HHO1* decreased by 98.12% and 97.60% in the shoots, respectively. In the roots, the expression levels of *BnaC9.HHO1* and *BnaA9.HHO1* was reduced by 98.62% and 99.55% under low NO_3_^−^ supply, respectively (Figure 8A). However, different *BnaGARPs* subfamilies showed distinct transcriptional responses under this circumstance. In detail, most (70%) of the differentially expressed genes (DEGs) of the *BnaGLKs* subfamily were upregulated in the shoots or roots under deficient NO_3_^−^ conditions (Figure 8B). In particular, the expression level of *BnaA6.GLK2* decreased by 60.37% in the roots, whereas the expression level of *BnaA2.GLK2* was increased 1.15-fold in the shoots. In terms of the *ARR* subfamily, the expression level of *BnaA3.APRR2* and *BnaC3.ARR1* was repressed by 53.55% and 66.89% in the roots of rapeseed plants exposed to deficient NO_3_^−^ conditions (Figure 8D). In the *BnaPHLs* subfamily, the expression level of *BnaAnn.PHL5* was increased 1.47-fold in the roots, whereas the expression level of *BnaC9.PHL1* was decreased by 79.90% in the shoots (Figure 8C).

To determine the core members that play a dominant role in the NO_3_^−^ response, a co-expression network analysis of *BnaGARPs* was performed. The results showed that *BnaA9.HHO1* and *BnaC7.HHO3* might play a major role in the repression of N-starvation responses in the shoots (Figure 8E); whereas in the roots, *BnaA9.HHO1* and *BnaC9.HHO1* might play a core role in the adaptation of rapeseed plants to N limitation (Figure 8F).

Under the ammonium (NH_4_^+^) supply condition, a total of 23 *BnaGARP* DEGs were identified in the shoots and roots relative to the condition of NO_3_^−^ sufficiency (Figure 9A). We found only the expression levels of *BnaA8.HHO5*, *BnaA1.HHO5*, and *BnaC3.HHO5* were increased by 1.51-fold, 1.24-fold, and 1.44-fold under the NH_4_^+^ supply condition than under NO_3_^−^ sufficiency. Among all the down-regulated genes, particularly the expression level of *BnaA6.HHO1* was reduced by 99.28% in the roots under the NH_4_^+^ supply condition.

Gene co-expression network analysis showed that *BnaA9.HHO1*, *BnaA7.HHO3* and *BnaA6.HHO1* might play a core role in the responses of rapeseed plants to NH_4_^+^ as the sole N nutrient source (Figure 9B,C).

Based on expression pattern analysis and co-expression network analysis, we selected several key genes and analyzed their differential expression between the high-NUE (H12) and low-NUE (L73) rapeseed cultivars (Figure 10A). The results showed that these genes were upregulated (1.53 to 6.64-fold) in the L73 rapeseed cultivar under NO_3_^−^ limitation condition (Figure 10B). In order to explore the role of these key genes involved in the regulation of differential NUE between the rapeseed genotypes, *BnaA9.HHO1* was selected to perform a functional analysis. The results showed that the *BnaA9. HHO1* fusion protein was mainly located in the nucleus and could colocalize with *OsGhd7* in the nucleus (Figure 10C).

Phosphate (Pi) performs a variety of biological functions, including structural elements in nucleic acids and phospholipids, signal transduction cascades, enzyme regulation, and so on [43]. Maeda et al., found that two independent transcriptional cascades for NO_3_^−^ and Pi-starvation signaling are integrated via expression control of the *GARP*-clade genes [10]. Under Pi limitation conditions, a total of 45 *BnaGARP* DEGs were identified in the shoots or roots (Figure 11). In the shoots, most of the DEGs were upregulated except for *BnaC7.GLK2*, *BnaA6.HHO6,* and *BnaC1.PHL2,* which were downregulated. In the *BnaNIGT1/HRS1/HHO* subfamily, *BnaC7.HHO1*, *BnaC8.HRS1b* and *BnaA9.HHO1* were remarkably upregulated, increasing by 6.78, 8.26, and 5.03-fold, respectively (Figure 11A). The expression levels of *BnaA7.ARR11* and *BnaA6.PCL1* had higher expression levels that were increased by 2.86-fold and 2.50-fold in the shoots under Pi deficiency than Pi sufficiency (Figure 11B,C). In terms of the *BnaPHLs* subfamily, the expression level of *BnaA9.PHL1* was decreased by 59.61%, while the expression level of *BnaC6.PHL8b* was increased by 1.02-fold in the roots under low Pi (Figure 11D).

### 2.8. Expression Profiles of BnaGARPs in Response to Diverse Nutrient Stresses

Further, the expression patterns of *BnaGARPs* under various nutrient stresses were studied, including deficient boron (B), salt stress, and cadmium (Cd) toxicity. The B requirement of plants varies from species to species, and *B. napus* is considered one of the highest B-requiring plants, which often suffers from yield and quality losses due to B deficiency, especially in Northern Europe, Canada, and China [44]. Under deficient B conditions, a total of 49 *BnaGARP* DEGs were identified in the shoots or roots. In the shoots, 39 DEGs were upregulated after B deficiency treatment (Figure 12). In particular, the expression level of *BnaAnn.PHL11* was increased 4.01-fold. In the *BnaARR-Bs* subfamily, the expression of three *BnaARRs* (including *BnaA2.ARR14*, *BnaA7.ARR11*, and *BnaC6.ARR11a*) was increased in the shoots after B deficiency treatment (Figure 12B). In the subfamily *BnaGLKs*, most of the genes had high expression levels (1.05 to 1.71-fold) under B deficiency than B sufficiency, whereas the expression of *BnaC1.PCL1* was reduced by 57.81% (Figure 12C). The expression pattern in subfamily *BnaNIGT1/HRS1/HHOs* and *BnaPHL1s* was similar to that in the subfamily of *BnaGLKs*. In the shoots, only *BnaA6.HHO6* and *BnaC1.PHL2* was downregulated (Figure 12D). Eight of 15 (53.33%) DEGs in the *BnaNIGT1/HRS1/HHO* subfamily and 15 of 18 (83.33%) DEGs in subfamily *BnaPHLs* were significantly induced by B deficiency.

Cd is a non-essential heavy metal with high biotoxicity to many organisms, while oilseed rape has a high potential for the phytoremediation of Cd-polluted soils [45]. Under Cd toxicity, a total of 43 *BnaGARP* DEGs were identified in the shoots or roots (Figure 12F). Most genes were downregulated in the roots in response to Cd toxicity, particularly the expression of *BnaC5.KAN2b* and *BnaCnn.PHL6* was reduced by 88.12% and 89.38%. In the shoots, Cd toxicity resulted in an obvious decrease in the expression of *BnaA1.PHL5*. Under Cd toxicity condition, *BnaA8.HHO5* was significantly increased by 2.92-fold in the roots.

Salt stress is one of the most important abiotic factors affecting global agricultural productivity, inhibiting plant growth, development and productivity by disrupting many physiological and biochemical processes [46]. In the salt stress-treated group, the expression of two *BnaARRs* (including *BnaA1.ARR2 and BnaA2.ARR14*) was induced by 1.53 and 1.14-fold in the shoots, whereas the expression of *BnaC3.ARR1* was significantly decreased (Figure 13). In the roots, *BnaA2.ARR14*, *BnaA3.APRR2,* and *BnaC7.APRR2* was upregulated by salt stress, while the expression of *BnaC3.ARR1* was decreased by 61.96% (Figure 13B). Under salt stress, *BnaA3.MYBC1*, *BnaA6.GLK2* and *BnaC9.MYBC1* in the *BnaGLK* subfamily showed a low expression level in both roots and shoots; however, *BnaA3.MYBC1b*, *BnaA5.MYBC1* and *BnaC4.MYBC1* in this subfamily shared higher expression levels under salt stress (Figure 13C). In terms of *BnaNIGT1/HRS1/HHO* subfamilies, most of them had lower expression levels (40.90% to 87.24%) under salt stress (Figure 13A). After salt treatment, the expression of 16 *BnaPHLs* was distinctly upregulated in the shoots, while the expression of eight *BnaPHLs* was obviously downregulated in the roots (Figure 13D).

To characterize the common genes responsive to nutrient stresses, a Venn diagram was constructed with the DEGs identified, respectively, under the diverse nutrient stresses above-mentioned. As shown in Figure 14, *BnaA9.HHO1* and *BnaA1.HHO5* was simultaneously regulated by low NO_3_^−^, NH_4_^+^ toxicity, limited Pi, deficient B, salt stress, and Cd toxicity in the shoots and roots (Figure 14). This result indicated that *BnaA9.HHO1* and *BnaA1.HHO5* might play a multifaceted role in regulating rapeseed resistance to nutrient stresses.

## 3. Discussion

Previous studies have shown that the *GARP* family members play critical roles in phytohormone transport and signaling, plant organ development, and nutrient responses [47,48,49]. However, there have been few systematic studies on *GARPs* in *B. napus*. In the present study, the genome-scale *GARP* family genes were identified in *B. napus* and their phylogenetic relationships, conserved motif and domain, gene structures, duplication and synteny relationships, CREs, and chromosomal locations were performed. In addition, we delineated the differential expression profile of *BnaGARPs* under low NO_3_^−^, NH_4_^+^ toxicity, limited Pi, deficient B, salt stress, and Cd toxicity. The global identification of *BnaGARPs* provides the foundation for further in-depth functional studies of these genes.

### 3.1. An Integrated Bioinformatics Analysis Provided Comprehensive Insights into the Molecular Features of BnaGARPs

In this study, a total of 146 *BnaGARPs* were identified (Appendix A). A previous study has revealed 56 *GARPs* in *A. thaliana*, 69 *GARPs* in *Camellia sinensis*, and 35 *GARPs* in *S. polyrhiza* [8,10,50], suggesting that the *GARP* TF family is ubiquitous in plants (Appendix A). The *GARP* gene family in rapeseed is larger than those in other plant species, which might be due to complex whole genome duplication and subsequent evolution of the rapeseed genome. [51]. Phylogenetic analysis showed that the *B. napus* genome retains the orthologs of *AtGARPs* and the gene phylogeny roughly followed the species phylogeny (Figure 1). Furthermore, the phylogenetic tree also showed that all subfamilies have expanded during the evolution process. A lineage-specific expansion of *BnaGARP* via the partial alteration of the genome is used to adapt to internal and external environments during evolution [52,53]. Generally, the Ka/Ks ratios for all the homologous *GARP* pairs were less than 1.0, indicating that *BnaGARPs* might have undergone purifying selection pressure (Appendix A). Arabidopsis and Brassica diverged about 20 million years ago, and evolutionary selection pressure analysis suggested that the divergence of *GARPs* also occurred during this period.

Due to the similarity between the B-motif and the MYB-like domain, the GARP TFs were frequently mistaken for MYB or MYB-like TFs. However, the MYB TFs contain the (SHAQK(Y/F) F) motif, while the *GARP* TFs contain a different consensus sequence (SHLQ (K/M) (Y/F)) [5]. All the *BnaGARPs* were predicted to contain some conserved motifs, which are components of the B-motif and are important for DNA binding (Figure 2). In this study, the conserved motifs in each subfamily of *BnaGARPs* are essentially similar, indicating that their amino acid residues are very conserved in terms of evolution, and have essential roles in gene function or structure. In addition, we found that *BnaNIGT1/HRS1/HHO* subfamily contains two different motifs EAR-like at their N or C terminal. The EAR-like motifs play an important role in inhibiting gene expression as transcription repressors or recruit corepressors [54]. In this study, different gene structures were found among *BnaGARPs*, and *BnaNIGT1/HRS1/HHO* subfamily had fewer exons than the *BnaPHL* subfamily, implying structural diversification among the *BnaGARP* subfamilies (Figure 3). The differences in the intron lengths suggested a possible role in the functional diversification of *BnaGARPs*. Chromosomal localization results showed that 146 genes are unevenly distributed on 20 chromosomes, presumably due to multiple polyploidization events in the genome of oilseed rape during its evolution [55]. Previous research revealed that tandem duplication events or segmental duplication events act as a mechanism for family expansion, and it also could promote the emergence of new functional genes that plants can better cope with abiotic stress during evolution [56,57].

To further elucidate the synteny relationships of *BnaGARPs* with *GARPs* in other model plants, we identified 172, 152, 66, 3, and 1 orthologous gene pairs between *BnaGARPs* with other *GARPs* in *G. max*, *A. thaliana*, *M. truncatula*, *O. sativa*, and *T. aestivum*, respectively (Figure 5). Synteny analysis results suggested that some *BnaGARPs* possibly came into being during gene duplication, and the segmental duplication events could play key roles in the expansion of *GARP* genes in *B. napus* [58]. In addition, *B. napus* and *A. thaliana* shared 152 syntenic gene pairs within the *GARP* family, indicating that *B. napus* and *A. thaliana* are closely evolutionarily related. Additionally, the allotetraploid *Brassica napus* L. (A_n_A_n_C_n_C_n_, 2n = 4x = 38) was formed by natural distant hybridization of diploid *Brassica rapa* L. (A_r_A_r_, 2n = 2x = 20) and diploid *Brassica oleracea* L. (C_o_C_o_, 2n = 2x = 18) [59]. In the present study, 45 duplication events took place on the A_n_ sub-genome, 36 events on the C_n_ sub-genome, and 136 events across A_n_/C_n_ sub-genomes. Therefore, we proposed that the *BnaGARP* expansion is a synergistic effect of polyploidization and hybridization working together [60].

The CREs in the promoter regions play an important role in regulating and functioning genes [61]. In this study, the CRE analysis confirmed the potential roles of *BnaGARPs* in the regulation of stress resistance (Figure 6). Many stresses and phytohormone-related CREs were identified in the promoter regions of most *GARPs*, including the ARE, G-box, MBS, and LTR elements. The most common CREs were light responsiveness CREs. Studies have confirmed that *AtHHO4* can interact with *JMJ30*, which is the *H3K36Me2* demethylase and is involved in light-responsive circadian clock [62].

MicroRNAs (miRNAs) are crucial non-coding regulators of gene expression in plants [63] and play essential roles in plant–environment interactions [64]. Over the past few years, a number of miRNAs have been recognized through genome-wide examination in rapeseed to participate in diverse nutrient stresses [38,65]. In this study, we identified 29 miRNAs targeting 34 *BnaGARPs* (Figure 7; Appendix A). miRNA164 has been reported to be involved in lateral root development in maize (*Zea mays* L.) [66]. miRNA156 has been reported to be significantly upregulated under dehydration stress responsiveness in different species [67]. Similarly, miR172 has also been found to regulate drought escape and drought tolerance by affecting sugar signaling in *A. thaliana* [68]. miR396 is a conserved miRNA and is involved in plant growth, development, and abiotic stress response in various plant species through regulating its targets, *Growth Regulating Factor* (*GRF*) TFs [69]. Some miRNAs have also been reported in rapeseed, playing a significant role in rapeseed genetic improvement [70,71]. These findings suggest that these bna-miRNAs might play pivotal roles against a variety of stresses by modifying the transcriptional or translational levels of *BnaGARPs*.

### 3.2. Differential Expression Profiling of BnaGARPs Implied Their Potential Involvement in the Responses of Rapeseed to Diverse Nutrient Stresses

TFs regulate gene expression by recognizing and combining *CREs* on the promoter regions of target genes [72]. TFs play key roles in plant developmental processes, phytohormone signaling pathways, and disease resistance responses. Given that expression patterns can lead to the estimation of gene functions [73]. For example, through analyzing the expression profile of *TaWRKY* family members under drought, cold, and high-temperature conditions, a considerable number of *TaWRKY* genes are shown to respond to drought stresses [3]. When exposed to ZnSO_4_ and FeCl_3_ solutions, the *TaZIP* genes showed differential expression patterns [74].

Previous studies have confirmed that *GARPs* play an important role in nutrient sensing [6]. The first *GARP* TF shown to be involved in nutritional responses was the Chlamydomonas phosphorus-stress response 1 (Psr1) [75]. Under Pi starvation, *OsPHR2* binds to a CRE (P1BS) in the promoter of various PSI genes and upregulates their transcription, thus optimizing rice Pi acquisition and utilization [76]. Another *GARP* subfamily that attracted recently lots of attention was *NIGT1/HRS1/HHO* subfamily. *NIGT1/HRS1/HHOs* have recently been confirmed to be involved in the perception and transduction of N and Pi nutritional signals in plant transcriptional regulatory networks [27].

In this study, we found that most *BnaGARPs* were significantly downregulated in the shoots or roots under NO_3_^−^ limitation conditions, among which the downregulated levels of *BnaNIGT1/HRS1/HHOs* were the highest (Figure 8). This finding highlighted the crucial role of *BnaNIGT1/HRS1/HHOs* in the regulation of NO_3_^−^ starvation. It has been demonstrated that *NIGT1*(*HHO1-HHO3*, *HRS1*) expression was induced by NO_3_^−^ signaling, and it also inhibited N starvation response (NSR) genes (*NRT2.1* and *NRT2.4*) under N sufficient conditions [10]. The *GARP* TFs modulate the expression of target genes by positive or negative feedforward mechanisms under abiotic stress [10]. For example, *AtNIGT1/HRS1* binds to the promoter of *NRT2.4* and represses an array of N starvation-responsive genes under conditions of high N availability [77]. *HRS1* and *HHO1* control ROS accumulation in response to NSR and directly repress NSR sentinel genes (*NRT2.5*) [78]. *NLPs* (including *NLP5* and *NLP7*) expression were downregulated by NLP-induced *NIGT1s* [10]. *SPX1*, *SPX2*, and *SPX4* are putative Pi-dependent inhibitors of Arabidopsis PHOSPHATE STARVATION RESPONSE1 (*PHR1*) [79]. To improve the understanding of *BnaGARP*-mediated transcriptional networks under abiotic stress responses, the transcriptional responses of 25 target genes were explored under these circumstances (Appendix A). Under NO_3_^−^ limitation conditions, *BnaAn.NRT2.4*, *BnaC9.NRT2.4*, and *BnaA8.NRT2.5* were upregulated, while *BnaA7.NLP5* and *BnaC6.NLP5* also shared higher expression levels. It indicates that *BnaNRT2.4* and *BnaNLP5* might play key roles in the *BnaGARP*-mediated transcriptional networks.

NH_4_^+^ is also a major N source for plants, and it is also an indispensable intermediate in the biosynthesis of essential cellular components [80]. In general, compared with NO_3_^−^, NH_4_^+^ as the sole N nutrient source had a weakened effect on the transcriptional responses of *BnaGARPs*. Under Pi limitation conditions, most of the *BnaGARP* DEGs were upregulated in the shoots or roots, among which the upregulated levels of *BnaC8.HRS1b* was the highest (Figure 11). Previous studies have reported Pi deprivation increased the *HRS1* expression level and expanded its expression domain [81]. Transcripts of *SPX1* and *SPX2* accumulate in the roots and shoots of Pi-limited plants in a PHR1-dependent manner [82]. In this study, *BnaSPX1* and *BnaSPX2* were upregulated in the roots and shoots under Pi-limited conditions. Moreover, we found no differences in the expression of *BnaKANs* under both NO_3_^−^ limitation and NH_4_^+^ toxicity conditions. A previous study has revealed that *AtKANs* regulate auxin biosynthesis, transport, and signaling [12]. Therefore, *BnaKANs* might be not involved in N absorption and utilization.

The expression patterns of *BnaGARPs* were also studied under various nutrient stresses. Under deficient B conditions, most *BnaGARP* DEGs were upregulated. Therefore, it could be concluded that *BnaGARPs* are also involved in response to B deficiency and might play important roles in B absorption in *B. napus* (Figure 12). Most members of the *BnaGARPs* have been shown to play a role in salt stress [18,83,84]. For instance, *HRS1* has transcriptional repressive activity and appears to suppress the expression of factors that negatively regulate salt tolerance, *ZmGLK3*, *SlGLK7*, and *SlGLK15* were upregulated under salt stress. In our results, we found that the expression level of *BnaHRS1* was significantly downregulated after salt stress (Figure 12). In the roots, *BnaA6.GLK2* and *BnaC7.GLK2* were upregulated. These results suggested that homologous genes should have similar expression patterns under abiotic stress. Moreover, most *BnaGARP* DEGs were downregulated under Cd toxicity (Figure 12F). It is worth noting that although some differential genes of *BnaGARPs* have been found, there are still some genes that have not been identified, which may be the problem of variety, expression site, and genome assembly.

In short, *BnaGARPs* were responsive to diverse nutrient stresses, which implied the essential roles of *BnaGARPs* in the resistance or adaptation of rapeseed to stresses.

### 3.3. BnaNIGT1/HRS1/HHOs Might Be Major Regulators of N-Starvation Responses

It has been reported that *NIGT1/HRS1/HHOs* were key regulators involved in plant response to limited NO_3_^−^ conditions. In Arabidopsis, the *NIGT1/HRS1/HHO* subfamily directly represses the expression of the *NRT2* genes (including *NRT2.1*, *NRT2.4*, and *NRT2.5*), *NLP* TFs directly activate genes encoding *NIGT1/HRS1/HHO* family TFs [10]. In rice, the overexpression of NIGT1 might have a negative effect on NUE and thus reduce the chlorophyll content [85]. In this study, *BnaNIGT1/HRS1/HHOs* were significantly downregulated under N-starvation responses (Figure 7). Among all the *BnaNIGT1/HRS1/HHO* DEGs, the transcription levels of *BnaA9.HHO1* and *BnaC9.HHO1* was most obviously downregulated. Furthermore, GFP-assisted subcellular localization analysis showed that *BnaA9.HHO1* was localized in the nucleus (Figure 9). Based on the co-expression network analysis and Venn diagram, we proposed that it was *BnaHHO1s* that might be the core genes in the N starvation response. However, functional validation is needed to reveal the in-depth functional roles of *BnaNIGT1/HRS1/HHOs*.

## 4. Materials and Methods

### 4.1. Identification of GARP Family Genes in Plants

In this study, the genomic, coding sequences, and protein sequences from *A. thaliana* and *B. napus* (Brana_ Dar_V5 genome) were downloaded from the Arabidopsis Information Resource (TAIR10, https://www.arabidopsis.org/, accessed on 1 October 2022) [86] and the Brassica Database (BRAD V3.0, http://brassicadb.cn/#/, accessed on 1 October 2022) [87]. To identify the *GARP* genes in these species, 56 *GARP* protein sequences from Arabidopsis were used as queries in a reciprocal Basic Local Alignment Search Tool (BLAST) analysis using the threshold and minimum alignment coverage parameters described previously [6,88]. All the *GARP* protein sequences were confirmed by comparison with *GARP* member sequences through searches of the Pfam (V35.0, http://pfam.xfam.org/, accessed on 1 October 2022) [89] and NCBI-CDD (https://www.ncbi.nlm.nih.gov/Structure/bwrpsb/bwrpsb.cgi, accessed on 1 October 2022) [90] database. The protein length, molecular weight (MW), and isoelectric point (pI) of each *GARP* protein were predicted using the ExPASy server (https://web.expasy.org/protparam/, accessed on 1 October 2022) [91].

The genes in Brassica species were named as follows: abbreviation of species name + chromosome + the name of gene homologs in *A. thaliana*. For example, *BnaC1.APRR2* represents a gene homologous to *APRR2* in *A. thaliana* on the C1 chromosome of *B. napus*.

### 4.2. Phylogenetic Analysis of the GARP Family in B. napus

Multiple sequence alignments of the *GARP* coding sequences between *B. napus* and *A. thaliana* were conducted using ClustalW2 (http://www.genome.jp/tools-bin/clustalw, accessed on 1 October 2022) [92] with default parameters. The phylogenetic trees were generated using the Molecular Evolutionary Genetics Analysis (MEGA) 7.0 program (https://megasoftware.net/home, accessed on 1 October 2022) [93] with the NJ method, the p-distance + G substitution model, 1000 bootstrap replications, and conserved sequences with a coverage of 70%. The phylogenetic trees were visualized using iTOL (V5, https://itol.embl.de/, accessed on 1 October 2022) [94]. The coding sequence alignments were imported into KaKs_calculator (https://ngdc.cncb.ac.cn/biocode/tools/BT000001, accessed on 1 October 2022) [95] to calculate the synonymous mutation rate (Ks) and non-synonymous mutation rate (Ka) using the NG method The time (T) of duplication in millions of years (Mya)was estimated with the formula T = Ks/2λ (λ = 1.5 × 10^−8^) [96].

### 4.3. Motif Identification and Gene Structure Analysis

Conserved motifs in the proteins were identified using the Expectation Maximization for Motif Elucidation program (MEME v4.12.0, https://meme-suite.org/meme/, accessed on 1 October 2022) [97] with the following parameter settings: the maximum number of motifs was 10. The conserved domains of *GARPs* were confirmed by NCBI-CDD search. TBtools was used to visualize the motifs and conserved domains of candidate genes. The gene structure was visualized by Gene Structure Display Server (2.0, http://gsds.gao-lab.org/, accessed on 1 October 2022) [98].

### 4.4. Chromosomal Locations and Synteny Analyses

Information about the physical locations of the *GARP* genes in the genomes of *B. napus* was collected from the BRAD database, and their positions were drafted to chromosomes by using MapGene2Chrom (http://mg2c.iask.in/mg2c_v2.1/, accessed on 1 October 2022) [99].

To uncover the evolutionary linear relationships within species and with ancestral species, the MCScanX plugin in TBtools V1.098 [100] was used to perform a collinearity analysis of *B. napus.* The circos plots of *BnaGARs* were generated by the Circos plugin in TBtools [101].

### 4.5. CRE Analysis

The CREs in the promoter regions of genes are considered to be related to the regulation of genes. In order to further investigate the potential regulatory network of *GARPs*, the 2000 bp upstream genomic DNA sequences of these genes’ start codon were submitted to PlantCARE (http://bioinformatics.psb.ugent.be/webtools/plantcare/html/, accessed on 1 October 2022) [102] to obtain CREs.

### 4.6. Prediction of Putative miRNA Targeting BnaGARPs

The cDNA sequences of *BnaGARPs* were used to identify possible target miRNAs in the psRNATarget database (V. 2017, Available online: https://www.zhaolab.org/psRNATarget/, accessed on 1 October 2022) [103] with default parameters, except maximum expectation (E) = 5.0. The targeted sites with high degrees of complementarity were selected. Cytoscape software (V3.8.2, https://cytoscape.org/download.html, accessed on 1 October 2022) [104] was used to create the interaction network between the prophesied miRNAs and the equivalent target BnaGARPs.

### 4.7. Plant Materials and Treatments

The *B. napus* seedlings (Darmor-bzh) germinated in this experiment. “Darmor-bzh” are a French winter oilseed rape variety, whose reference genome sequence was first published in 2014 [59].

First, plump *B. napus* seeds were selected, disinfected with 1% NaClO for 10 min, cleaned with ultra-pure water, soaked overnight at 4 °C, and then sown on the seedling tray. The 7-d old uniform *B. napus* seedlings after seed germination were transplanted into black plastic containers with 10 L Hoagland nutrient solution. The basic nutrition solution contained 1.0 mM KH_2_PO_4_, 5.0 mM KNO_3_, 5.0 mM Ca(NO_3_)_2_·4H_2_O, 2.0 mM MgSO_4_·7H_2_O, 0.050 mM EDTA-Fe, 9.0 μM MnCl_2_·4H_2_O, 0.80 μM ZnSO_4_·7H_2_O, 0.30 μM CuSO_4_·5H_2_O, 0.10 μM Na_2_MoO_4_·2H_2_O, and 46 μM H_3_BO_3_. The rapeseed seedlings were cultivated in an illuminated chamber following the growth regimes: light intensity of 300–320 μmol m^−2^ s^−1^, temperature of 25 °C daytime/22 °C night, light period of 16 h photoperiod/8 h dark, and relative humidity of 70% [105].

To further analyze the expression patterns of *BnaGARPs* under different nutrient stresses, five treatments were set. For the NO_3_^−^ depletion treatment, the 7-d old uniform *B. napus* seedlings were hydroponically cultivated under high (6.0 mM) NO_3_^−^ for 10 d, and then were grown under low (0.30 mM) NO_3_^−^ for 3 d until sampling. For the NH_4_^+^ toxicity treatment, the 7-d-old uniform *B. napus* seedlings after seed germination were hydroponically cultivated under high NO_3_^−^ for 10 d and then were grown under N-free conditions for 3 d. Finally, the plants were grown under excess (9.0 mM) NH_4_^+^ for 6 h until sampling. For the inorganic Pi starvation treatment, the 7-d-old uniform *B. napus* seedlings after seed germination were first hydroponically grown under 250 μM Pi (KH_2_PO_4_) for 10 d, and then were grown under 5 μM Pi for 3 d until sampling. For the salt stress treatment, the 7-d-old uniform *B. napus* seedlings after seed germination were hydroponically cultivated in a NaCl-free solution for 10 d and then were transferred to 200 mM NaCl for 1 d until sampling. In the B deficiency treatment, the 7-d-old uniform *B. napus* seedlings after seed germination were first hydroponically grown under 10 μM H_3_BO_3_ for 10 d, and then were transferred to 0.25 μM H_3_BO_3_ for 3 d until sampling. For the Cd toxicity treatment, the 7-d-old uniform *B. napus* seedlings after seed germination were hydroponically cultivated in a Cd-free solution for 10 d and then were transferred to 10 μM CdCl_2_ for 12 h until sampling. In addition, a high-NUE (H73) and a low-NUE (L12) rapeseed cultivar were also used for the experiment under nitrate limitation conditions [106].

The shoots and roots of fresh rapeseed seedlings above-mentioned were sampled separately and were immediately stored at 80 °C. Each sample contained three independent biological replicates for the transcriptional analyses of *BnaGARPs* under diverse nutrient stresses.

### 4.8. Transcriptional Analysis of BnaGARPs under Diverse Nutrient Stresses

A total of 12 RNA samples from each treatment were subjected to an Illumina HiSeq X Ten platform (Illumina Inc., San Diego, CA, USA). The illumine RNA-seq data were analyzed to reveal the transcriptional responses of *BnaGARPs* under diverse nutrient stresses. To identify the DEGs between different samples/groups, the expression level of each gene was calculated according to the TPM method. RSEM (http://deweylab.biostat.wisc.edu/rsem/, accessed on 1 October 2022) [107] was used to quantify gene abundances. Essentially, differential expression analysis was performed using DESeq2 [108], and the DEGs with |log_2_ (FC)| ≥ 1 and P-adjust ≤ 0.05.

### 4.9. Subcellular Localization Assay

Subcellular localization of target genes was determined using polyethylene glycol-mediated protoplast transformation in Arabidopsis [109]. *OsGhd7* was used as a nuclear marker and fused with a red fluorescent protein sequence [110]. Fluorescence was observed using a Nikon C2-ER confocal laser-scanning microscope with emission filters set at 510 nm (GFP) and 580 nm (RFP), and excitation was achieved at 488 nm (GFP) and 561 nm (RFP).

## 5. Conclusions

In this study, a systematic genome-wide analysis and molecular characterization of the 146 *GARP* members in allotetraploid rapeseed was completed. In addition, RNA-seq data showed that *BnaGARPs* respond to various nutritional stresses Among all DEGs, *BnaA9.HHO1* and *BnaA1.HHO5* might play a core role in regulating rapeseed resistance to nutrient stresses. However, additional investigations are required to confirm the functional roles of these core genes. The present results would increase the understanding of the evolution of the *GARP* family genes and provide valuable candidate genes for further study of the transcriptional regulation mechanism in response to nutrient stresses in rapeseed.

## Figures and Tables

**Figure 1 ijms-23-14484-f001:**
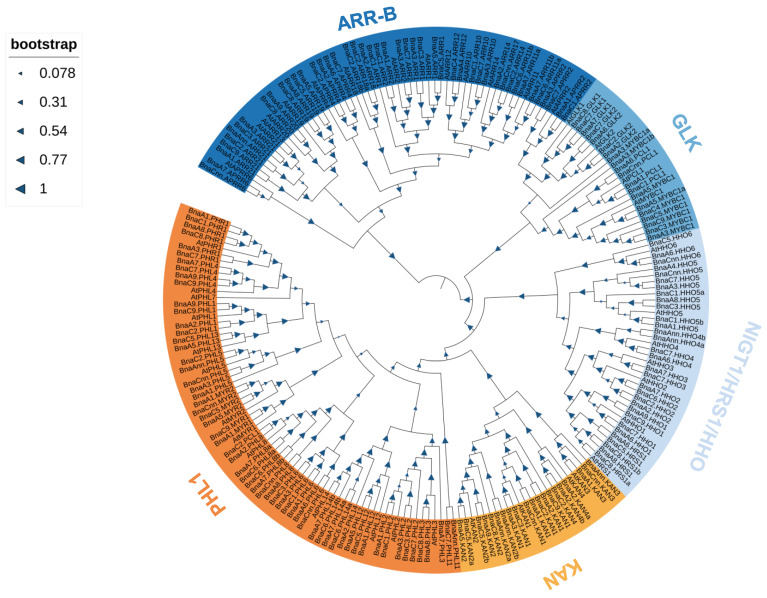
Phylogenetic tree of the *GARPs* retrieved from *B. napus* and *A. thaliana*. The phylogenetic tree was constructed according to the neighbor-joining method. The tree was generated using MEGA7.0 based on the *GARP* amino acid sequences retrieved from *B. napus* and *A. thaliana*. The genes from each group are indicated by different colors. The rectangle sizes at the nodes represent the bootstrap values.

**Figure 2 ijms-23-14484-f002:**
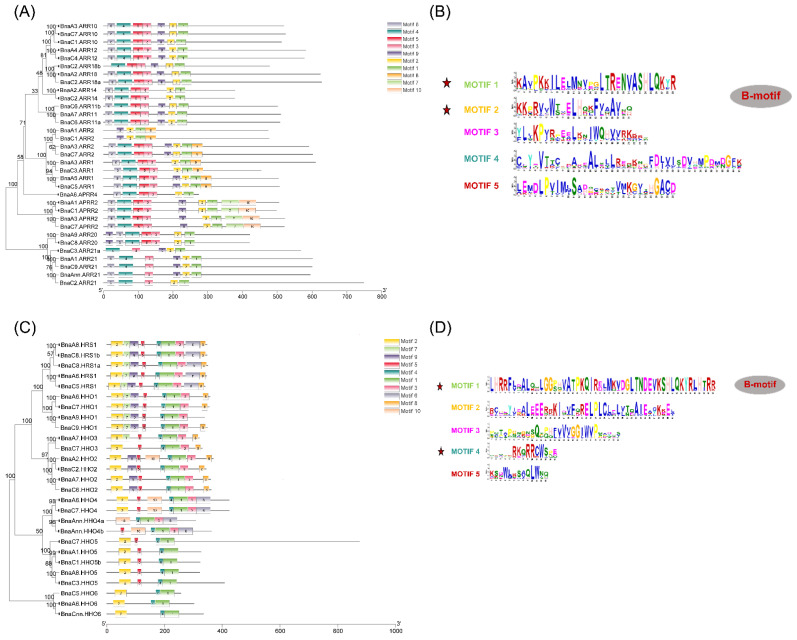
Identification and characterization of the conserved motifs in the GARP proteins in *B. napus*. (**A**) Molecular identification of *BnaARR-Bs*. (**B**) The sequence characterization of *BnaARR-Bs*. (**C**) Molecular identification of *BnaNIGT1/HRS1/HHOs*. (**D**) The sequence characterization of *BnaNIGT1/HRS1/HHOs*. In A and C, the boxes with different colors indicate different conserved motifs (motifs 1–10), and black lines represent the GARP protein regions without detected motifs. In C and D, the larger the fonts, the more conserved the motifs. Among them, the tagged motifs were identified as the B-motifs.

**Figure 3 ijms-23-14484-f003:**
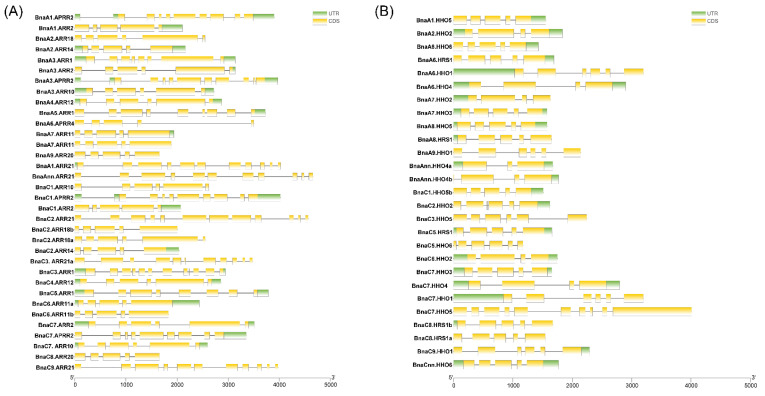
Exon-intron organizations of *BnaNIGT1/HRS1/HHOs* (**A**) and *BnaARR-Bs* (**B**). The green boxes represent untranslated regions, the yellow boxes represent exons, and the black lines represent the introns. The lengths of the exons and introns can be determined by the scale at the bottom.

**Figure 4 ijms-23-14484-f004:**
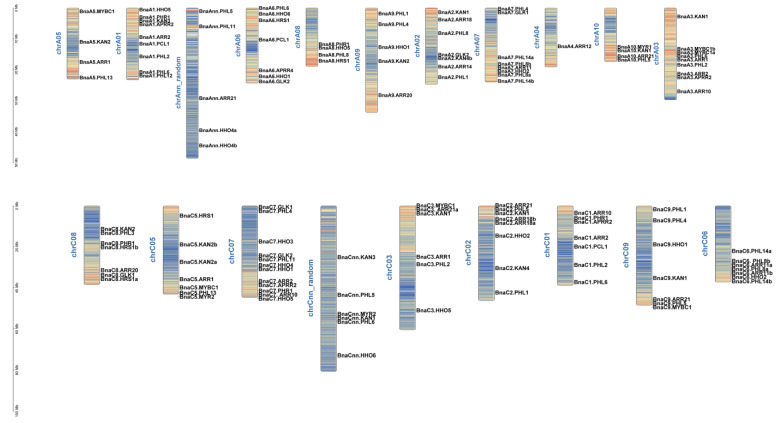
Chromosomal location of the 146 genes in *GARP* family genes in *B. napus*. The distribution of the 146 genes on the 20 chromosomes is presented. The Ann and Cnn chromosomes refer to the chromosome that is anchored to the A and C subgenomes, while they have been not the specific chromosome.

**Figure 5 ijms-23-14484-f005:**
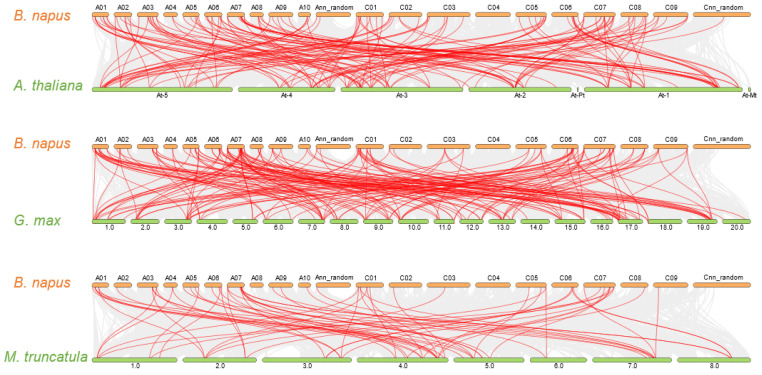
Synteny analysis of the *GARP* family genes between *B. napus* and *A. thaliana*, *G. max*, and *M. truncatula*. Gray lines indicate all collinear blocks within *B. napus* and *A. thaliana*, *G. max*, and *M. truncatula*. While the red lines depict the orthologous relationships.

**Figure 6 ijms-23-14484-f006:**
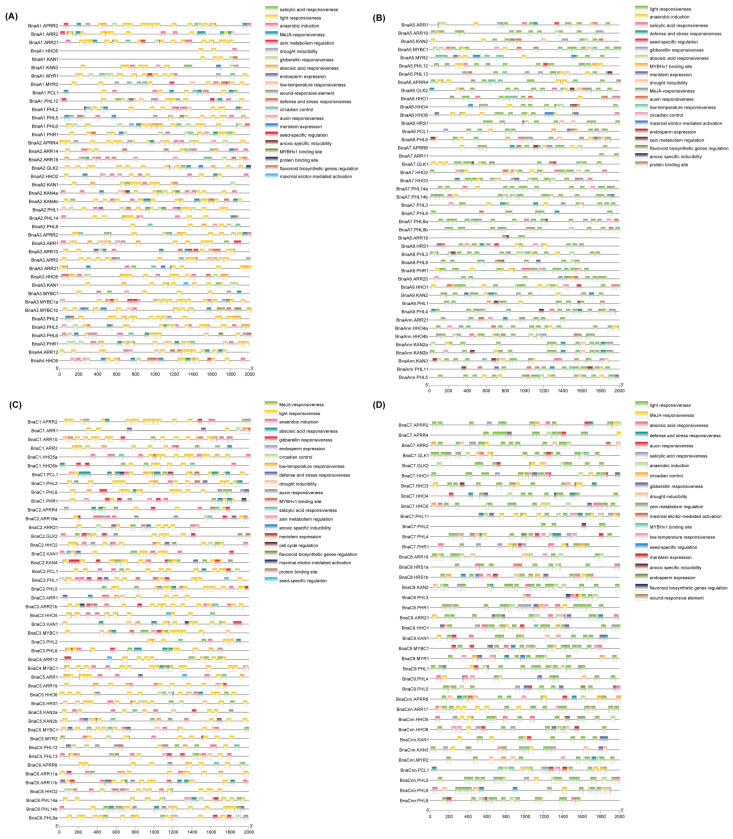
Predicted *cis*-regulatory elements (CREs) in the promoter regions of *BnaGARPs*. (**A**) Distribution of the CREs on chrA1 to chrA4. (**B**) Distribution of the CREs on chrA05 to chrAnn. (**C**) Distribution of the CREs on chrC1 to chrC6. (**D**) Distribution of the CREs on chrC7 to chrCnn. The CREs identified by PlantCARE are based on the sequence of 2000 bp upstream of the start codon of *BnaGARPs*. Different colored rectangles represent different CREs that are potentially involved in the regulation of stress resistance or phytohormone response.

**Figure 7 ijms-23-14484-f007:**
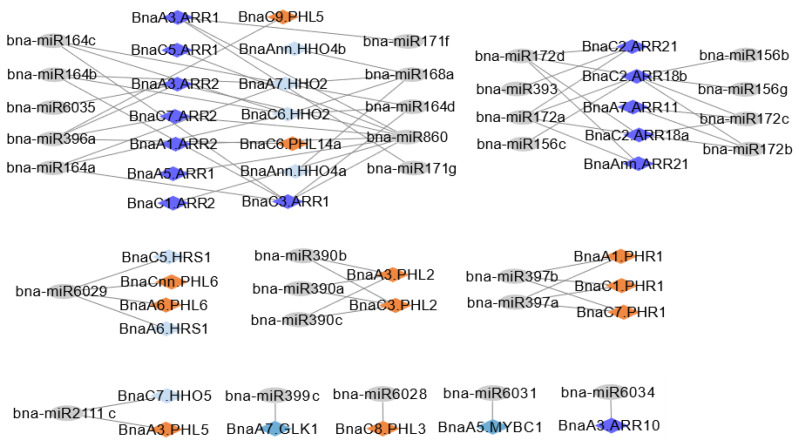
Network diagrams of predicted miRNAs targeting *BnaGARPs*. Different diamond colors represent *BnaGARPs*, and gray ellipse shapes represent potential regulatory miRNAs.

**Figure 8 ijms-23-14484-f008:**
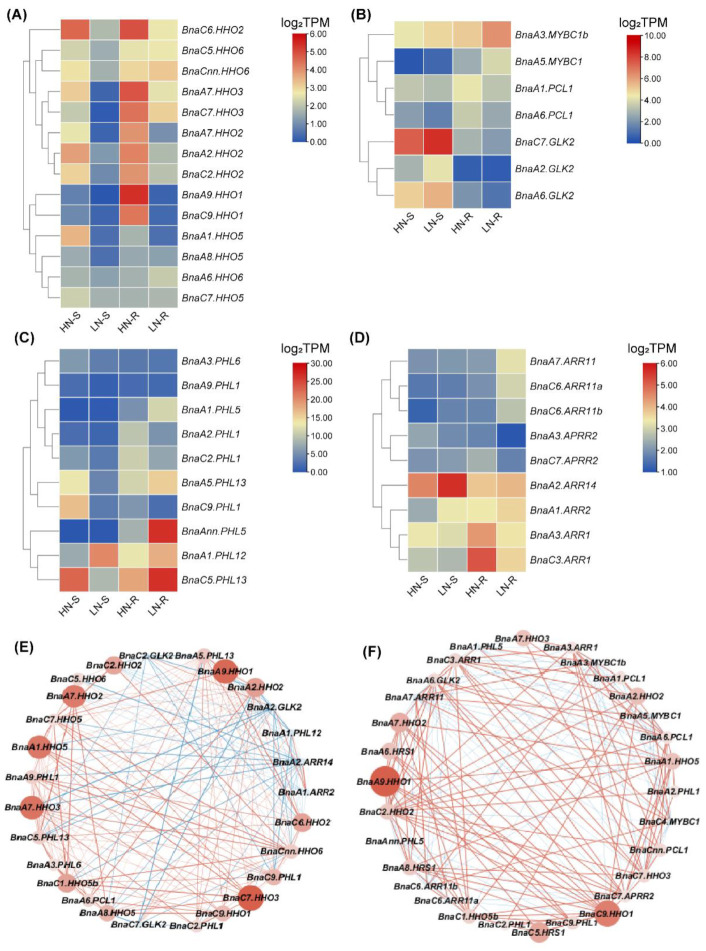
Expression profiles and co-expression network analysis of *BnaNIGT1/HRS1/HHOs* (**A**), *BnaARRs* (**B**), *BnaGLKs* (**C**), and *BnaPHLs* (**D**) in the shoots/S (**E**) and roots/R (**F**) under nitrate (NO_3_^−^) limitation conditions. HN, high N (6.0 mM NO_3_^−^); LN, low N (0.30 mM NO_3_^−^). In the heat maps, the expression levels are normalized by log_2_ (TPM). TPM, transcripts per million (reads). The color scales represent relative expression levels from high (red color) to low (blue color). In the gene co-expression networks, the cycle nodes represent genes, and the size of the nodes represents the power of the interrelation among the nodes by log_2_FC value. FC, fold change. The edges between two nodes represent interactions between genes.

**Figure 9 ijms-23-14484-f009:**
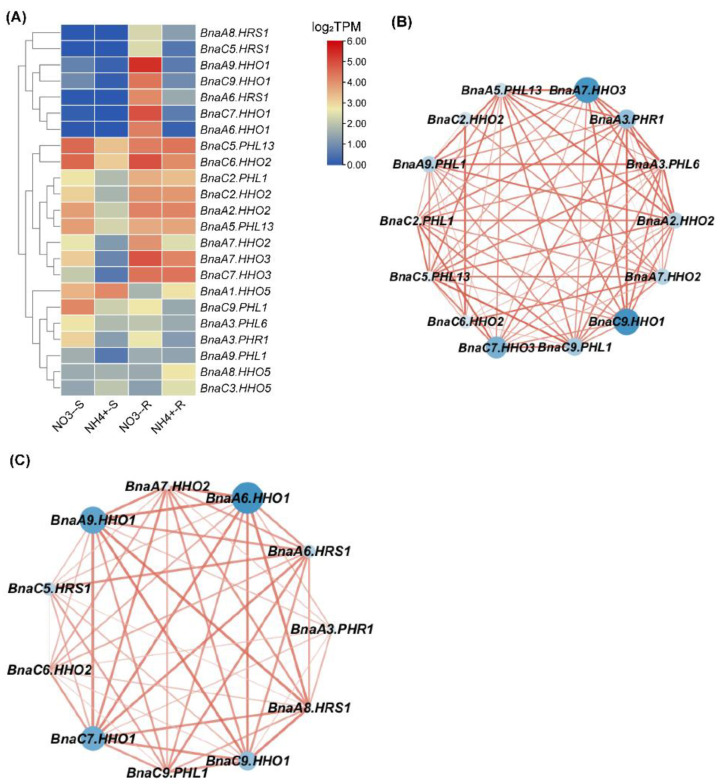
Expression profiles (**A**) and co-expression network analysis of *BnaGARPs* in the shoots/S (**B**) and roots/R (**C**) under different nitrogen (N) form conditions, including 6.0 mM nitrate (NO_3_^−^) and 6.0 mM ammonium (NH_4_^+^) conditions. The expression levels are normalized by log_2_(TPM). TPM, transcripts per million (reads). In the heat maps, the color scales represent relative expression levels from high (red color) to low (blue color). In the gene co-expression networks, the cycle nodes represent genes, and the size of the nodes represents the power of the interrelation among the nodes by log_2_FC value. FC, fold change. The edges between two nodes represent interactions between genes.

**Figure 10 ijms-23-14484-f010:**
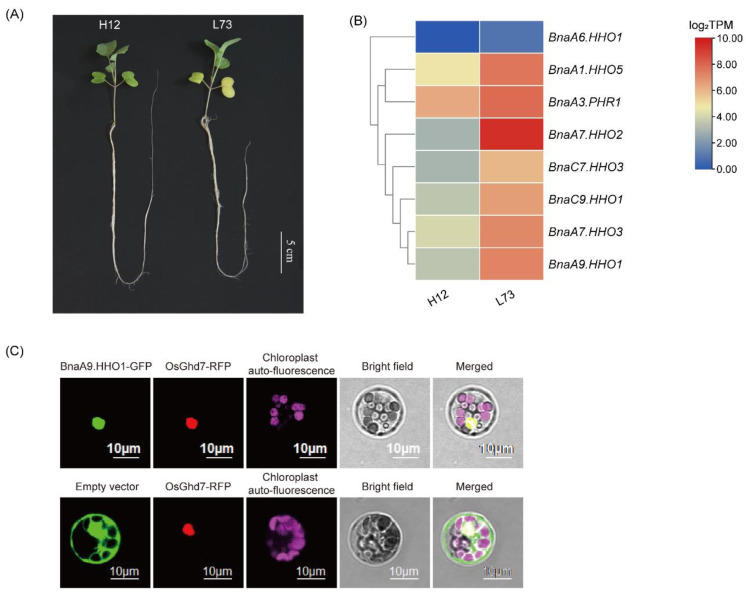
Characterization of the rapeseed genotypes H12 and L73 growing under nitrate (NO_3_^−^) limitation conditions and subcellular localization analysis of *BnaA9.HHO1*. (**A**) Growth performance of the high-NUE genotype H12 and the low-NUE genotype L73 growing under low (0.3 mM) NO_3_^−^ condition. (**B**) Expression profiles of the *BnaGARP* DEGs between H12 and L73 under NO_3_^−^ limitation conditions. (**C**) Subcellular localization analysis of *BnaA9.HHO1*. *OsGhd7* was used as a nuclear marker and fused with a red fluorescent protein sequence.

**Figure 11 ijms-23-14484-f011:**
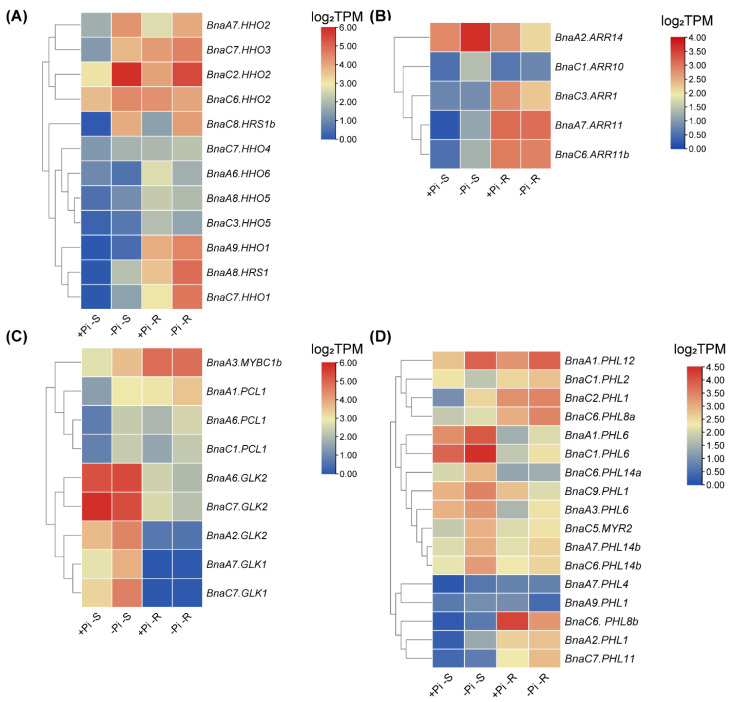
Expression profiles of *BnaNIGT1/HRS1/HHOs* (**A**), *BnaARRs* (**B**), *BnaGLKs* (**C**), and *BnaPHLs* (**D**) in the shoots/S and roots/R under different phosphate (Pi) levels. conditions. +Pi, high Pi (250 μM); -Pi, low Pi (5 μM), The expression levels are normalized by log_2_ (TPM). TPM, transcripts per million (reads). The color scales represent relative expression levels from high (red color) to low (blue color).

**Figure 12 ijms-23-14484-f012:**
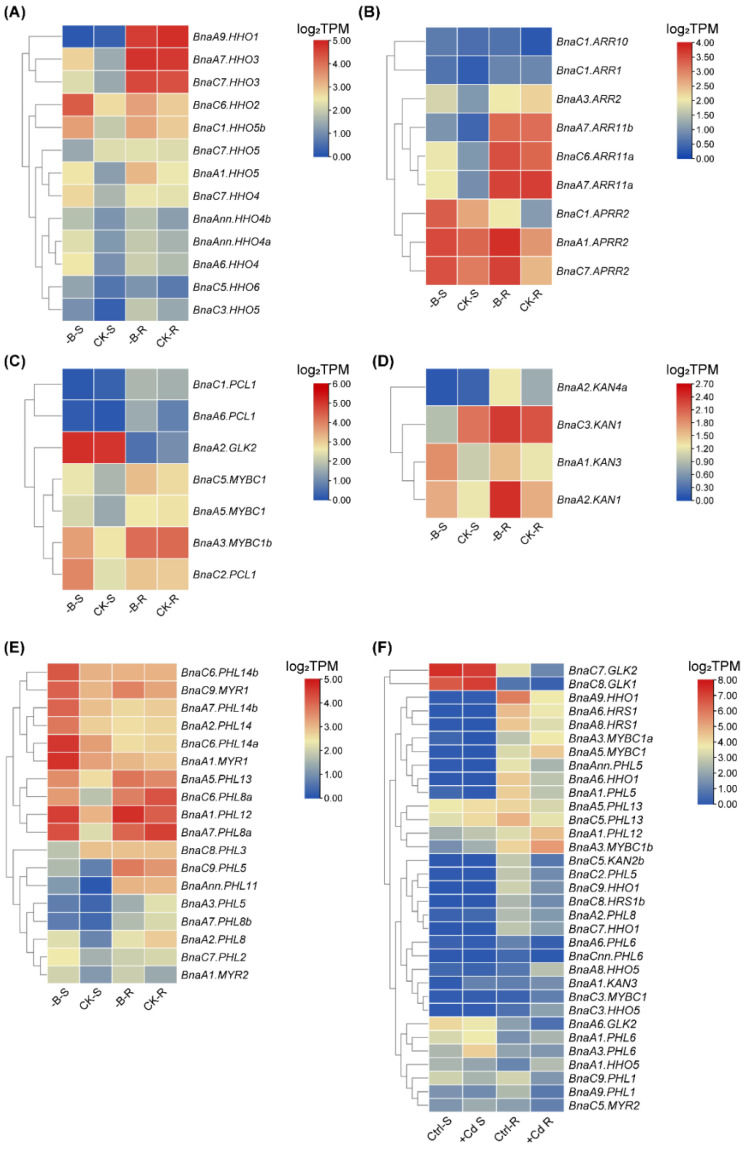
Expression profiles of *BnaNIGT1/HRS1/HHOs* (**A**), *BnaARRs* (**B**), *BnaGLKs* (**C**), *BnaKANs* (**D**), and *BnaPHLs* (**E**) in the shoots/S and roots/R under different boron (**B**) and cadmium (Cd) toxicity (**F**) conditions. -B, low B (0.25 μM); CK, high B (25 μM); Ctrl, Cd-free; +Cd, high Cd (10 μM). In the heat maps, the expression levels are normalized by log_2_ (TPM). TPM, transcripts per million (reads). The color scales represent relative expression levels from high (red color) to low (blue color).

**Figure 13 ijms-23-14484-f013:**
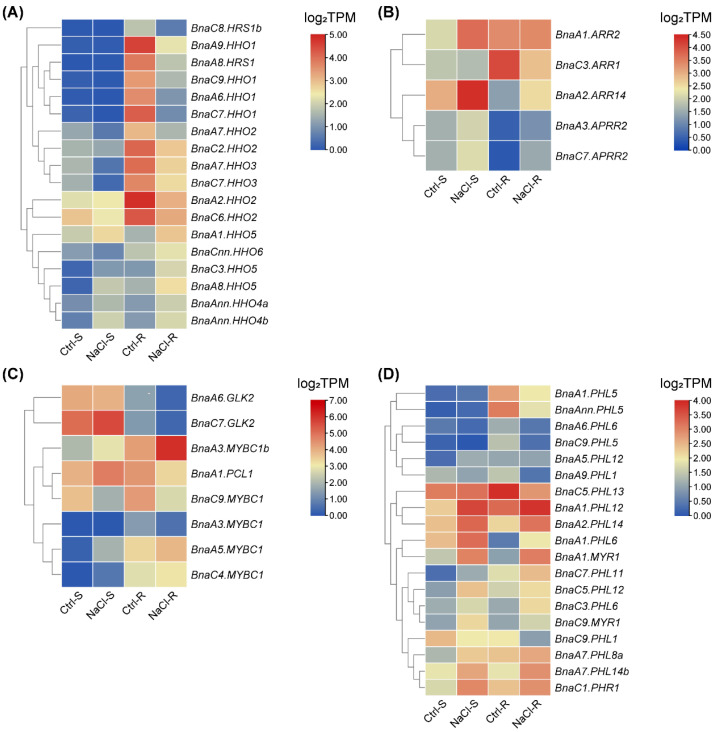
Expression profiles of *BnaNIGT1/HRS1/HHOs* (**A**), *BnaARRs* (**B**), *BnaGLKs* (**C**), and *BnaPHLs* (**D**) in the shoots/S and roots/R under salt stress conditions. Ctrl, control, NaCl-free; NaCl, +NaCl, 200 mM. In the heat maps, the expression levels are normalized by log_2_ (TPM). TPM, transcripts per million (reads). The color scales represent relative expression levels from high (red color) to low (blue color).

**Figure 14 ijms-23-14484-f014:**
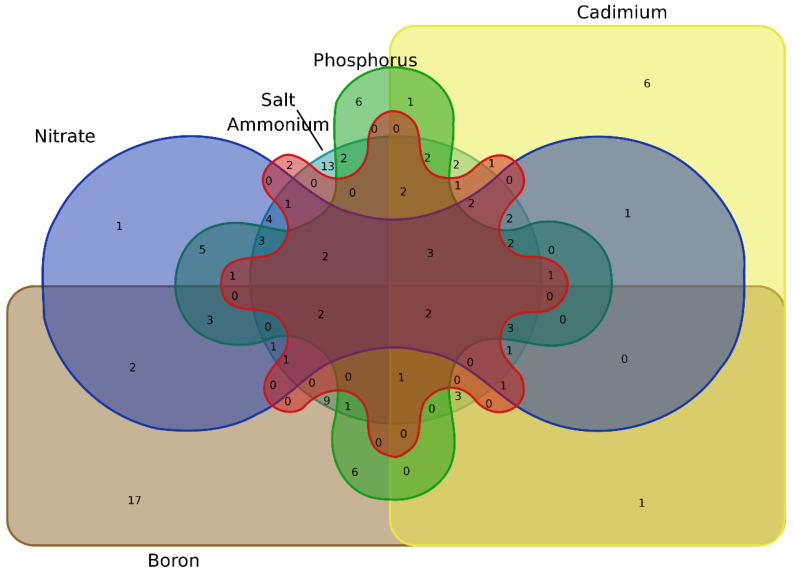
Venn diagram showing the common and specific differentially expressed genes of *BnaGARPs* under diverse nutrient stresses.

## Data Availability

All the data and plant materials in relation to this work can be obtained through contacting with the corresponding author Dr. Ting Zhou (zhoutt@zzu.edu.cn).

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
