# Peer review of "Genome-Scale Investigation of GARP Family Genes Reveals Their Pivotal Roles in Nutrient Stress Resistance in Allotetraploid Rapeseed"

_ijms, 2022, doi:10.3390/ijms232214484_

Round 1
Reviewer 1 Report
This study performed a systematic genome-wide analysis and molecular characterization of the GARP gene family members in Brassica napus. A total of 146 GARP genes were identified in B.napus, which is the highest number found for any species previously studied. In addition, RNA-seq data showed that BnaGARPs respond to various nutritional stresses, especially nitrogen starvation conditions. Overall, the study is nicely designed and executed. Still, there is a scope for further improvement. Thus, I suggest major revision.
1. Line 16, also add common name like rapeseed (Brassica napus L.).
2. Line 24, kindly add some potential genes within parenthesis.
3. Revise all the keywords. The keywords should not be similar to the title.
4. Line 88, ref 27 is wrong as it deals with NIGT1 family proteins in Arabidopsis. Authors must cite a most suitable reference for rapeseed, such as DOI: 10.1007/s00344-020-10231-z.
5. Line 87-88, write a few lines about how different abiotic stresses like drought, salinity, and temperature affect rapeseed production. Here is a suggestion for a suitable paper DOI: 10.1007/s00344-020-10231-z.
6. All scientific names must appear in italics, such as line 107.
7. In some figures, the text is too short and not clear. Kindly improve the quality of the figures.
8. In all expression-related parts, I suggest to discuss the results with numerical values such as FC or % for up- or down-regulated genes at particular stress and cultivar.
9. Though the current data is also enough to warn the publication. I think the prediction of putative miRNAs targeting GARP members would be a nice addition. Thus, I suggest authors to prediction the putative miRNAs, for instance, https://www.mdpi.com/2076-3921/10/8/1182; https://doi.org/10.1186/s12864-021-07862-1. In these rapeseed studies, authors have identified several putative miRNAs targeting candidate genes. A similar approach can be adopted for GARP members.
10. Line 335, ref 34 should be replaced with a recent review DOI: 10.1080/07388551.2022.2093695.
11. Disucssion does not cover the whole study. Kindly improve the discussion. Mainly, the expression parts need to be properly discussed with recent literature.
12. In conclusion, I suggest highlighting some candidate genes for future functional analysis. Also, add a few future recommendations.
13. In one supplementary excel sheet, I suggest authors provide all the gene's names, original IDs, and protein sequences.
Author Response
Point-to-point response to Reviewer 1#
General comment: This study performed a systematic genome-wide analysis and molecular characterization of the GARP gene family members in Brassica napus. A total of 146 GARP genes were identified in B.napus, which is the highest number found for any species previously studied. In addition, RNA-seq data showed that BnaGARPs respond to various nutritional stresses, especially nitrogen starvation conditions. Overall, the study is nicely designed and executed. Still, there is a scope for further improvement. Thus, I suggest major revision.
Response: We appreciate for your warm work and thanks very much for your positive and constructive comments on our manuscript. Your comments are considerably valuable and very helpful for revising and improving our paper, particularly on the experimental design for proceeding the project in the future. We have studied your comments carefully and have made corrections in red with tracked changes which we hope meet with your approval. The point-by-point responses to your comments are also listed as below.
Comment 1: Line 16, also add common name like rapeseed (Brassica napus L.).
Response: Thank you very much for your helpful comments. In view of this valuable suggestion, we have added the common name in the Abstract of the revised manuscript, which is as follows:
…
However, few systematic analyses of GARPs have been reported in allotetraploid rapeseed (Brassica napus L.) yet. (Page 1, lines15-16)
Comment 2: Line 24, kindly add some potential genes within parenthesis.
Response: Thank you very much for your kind suggestion. According to your advice, we have added this part in Abstract, which is as follows:
…
Notably, BnaA9.HHO1 and BnaA1.HHO5 were simultaneously transcriptionally responsive to these nutrient stresses in both hoots and roots, which indicated that BnaA9.HHO1 and BnaA1.HHO5 might play a core role in regulating rapeseed resistance to nutrient stresses. (Page 1, lines 26-29)
Comment 3: Revise all the keywords. The keywords should not be similar to the title.
Response: Thank you very much for your kind suggestion. According to your advice, we have revised the keywords, which is as follows:
Keywords: Brassica napus; Transcription factors; Nutrient stress; Transcriptomic analysis; miRNA (Page 1, line 32)
Comment 4: Line 88, ref 27 is wrong as it deals with NIGT1 family proteins in Arabidopsis. Authors must cite a most suitable reference for rapeseed, such as DOI: 10.1007/s00344-020-10231-z.
Response: Thank you very much for your kind suggestion. According to your advice, we have corrected the reference in our latest version, which as follows:
…
Rapeseed (Brassica napus L.) is a major oilseed crop due to its economic value and oilseed production. However, its productivity has been reduced by many environmental adversities [30]. (Page 2, lines 94-97)
30 Raza, A.; Razzaq, A.; Mehmood, S.S.; Hussain, M.A.; Wei, S.; He, H.; Zaman, Q.U.; Xuekun, Z.; Hasanuzzaman, M. Omics: The way forward to enhance abiotic stress tolerance in Brassica napus L. GM Crops Food. 2021, 12(1), 251-281. doi: 10.1080/21645698.2020.1859898.
Comment 5: Line 87-88, write a few lines about how different abiotic stresses like drought, salinity, and temperature affect rapeseed production. Here is a suggestion for a suitable paper DOI: 10.1007/s00344-020-10231-z.
Response: Thank you very much for your kind suggestion. In view of this valuable suggestion, we have added this part to the Introduction of the revised manuscript, which is as follows:
…
Under drought tolerance, the shoot and root growth of rapeseed seedlings is greatly inhibited, which ultimately will reduce crop production [31]. For rapeseed, salt stress severely affects all life stages from seed germination to yield production [32]. Cold stress has a negative impact on rapeseed germination and seedling establishment, causing wilting and plant death at the seedling stage [33]. Rapeseed production in the field is also often severely inhibited due to N deficiency [34], and is highly dependent on N fertilizer application, but its N use efficiency (NUE) is very low [35]. Rapeseed is also extremely sensitive to Pi deficiency [36]. A number of gene families, such as superox-ide dismutase (SOD) [37], lipid phosphate phosphatases (LPP) [38], B-box (BBX) [39] play critical roles in rapeseed growth, development, and response to stresses in rape-seed. (Pages 2-3, lines 97-106)
Comment 6: All scientific names must appear in italics, such as line 107.
Response: Thank you very much for your kind suggestion. According to your advice, we have corrected these mistakes you mentioned, which is as follows:
…
the sequences of 146 B. napus GARP proteins, and 56 A. thaliana GARP proteins were used to construct a phylogenetic tree. (Page 3, lines 135-136)
…
Moreover, the number of GARPs in B. napus is much more than three times of those in A. thaliana. (Page 3, lines 137-138)
Comment 7: In some figures, the text is too short and not clear. Kindly improve the quality of the figures.
Response: Thank you very much for your kind suggestion. According to your advice, we have improved the quality of the figures in the Figure zip file that we submitted.
Comment 8: In all expression-related parts, I suggest to discuss the results with numerical values such as FC or % for up- or down-regulated genes at particular stress and cultivar.
Response: Thank you very much for your kind suggestion. In view of this valuable suggestion, we have modified the expression-related parts to the Results of the revised manuscript, for example:
…
In BnaNIGT1/HRS1/HHOs subfamily, most members were downregulated (87.88%-98.12%) in the shoots under the low NO3- supply. Notably, the expression levels of BnaC7.HHO3 and BnaA9.HHO1 were decreased by 98.12% and 97.60% in the shoots respectively. (Page 10, lines 267-270)
…
In particular, the expression level of BnaA6.GLK2 decreased by 60.37% in the roots, whereas the expression level of BnaA2.GLK2 was increased 1.15-fold in the shoots. (Page 10, lines 275-277)
…
In the BnaPHLs subfamily, the expression level of BnaAnn.PHL5 was increased 1.47-fold in the roots, whereas the expression level of BnaC9.PHL1 was decreased by 79.90% in the shoots (Page 10, lines 279-281)
Comment 9: Though the current data is also enough to warn the publication. I think the prediction of putative miRNAs targeting GARP members would be a nice addition. Thus, I suggest authors to prediction the putative miRNAs, for instance, https://www.mdpi.com/2076-3921/10/8/1182; https://doi.org/10.1186/s12864-021-07862-1. In these rapeseed studies, authors have identified several putative miRNAs targeting candidate genes. A similar approach can be adopted for GARP members.
Response: Thank you very much for your kind suggestion. According to your suggestion, we have added the prediction of putative miRNAs targeting GARP members in the Results and Discussion
section of the revised manuscript, which is also as follows:
Results
…
2.6 Genome-Wide Analysis of miRNA Targeting BnaGARPs
In plants, miRNAs are important regulators of gene expression and play pivotal roles in abiotic stress responses [41]. To identify whether miRNAs are involved in the regulation of the BnaGARP expression, we identified 29 putative miRNAs targeting 34 BnaGARPs (Figure 7). Some of the miRNA-targeted sites are presented in Figure S3, while the detailed information of all miRNAs targeted genes is presented in Table S10. The results showed that four members of the bna-miR164 family targeted three Bna-GARPs (including BnaC3.ARR1, BnaC6.HHO2, and BnaA7.HHO2). Four members of the bna-miR172 family targeted two BnaGARPs (including BnaC2.ARR18a and BnaC2.ARR18b). (Page 9, lines 241-257)
Discussion
…
MicroRNAs (miRNAs) are crucial non-coding regulators of gene expression in plants [64], and play essential roles in plant–environment interactions [65]. Over the past few years, a number of miRNAs have been recognized through genome-wide ex-amination in rapeseed to participate in diverse nutrient stresses [38, 66]. In this study, we identified 29 miRNAs targeting 34 BnaGARPs (Figure 7; Table S4-S8). miRNA164 has been reported to be involved in lateral root development in maize (Zea mays L.) [67]. (Pages 18-19, lines 478-492)
Comment 10: Line 335, ref 34 should be replaced with a recent review DOI: 10.1080/07388551.2022.2093695.
Response: Thank you very much for your kind suggestion. We further replace this reference in our latest version, which is as follows:
Salt stress is the second most important abiotic factor affecting global agricultural productivity, affecting plant growth, development and productivity by disrupting many physiological and biochemical processes [46].
46 Raza, A.; Tabassum, J.; Fakhar, A.Z.; Sharif, R.; Chen, H.; Zhang, C.; Ju, L.; Fotopoulos, V; Siddique, K.H.M; Singh, R.K; Zhuang, W.; Varshney, R.K.; Smart reprograming of plants against salinity stress using modern biotechnological tools. Crit Rev Biotechnol. 2022, 15, 1-28. doi: 10.1080/07388551.2022.2093695.
Comment 11: Disucssion does not cover the whole study. Kindly improve the discussion. Mainly, the expression parts need to be properly discussed with recent literature.
Response: Thank you very much for your kind suggestion. In view of this valuable suggestion, we have readjusted the structure and content of the Discussion in the revised manuscript.
Comment 12: In conclusion, I suggest highlighting some candidate genes for future functional analysis. Also, add a few future recommendations.
Response: Thank you very much for your kind suggestion. In view of this valuable suggestion, we have added this part to the conclusion of the revised manuscript, which is as follows:
…
Among all DEGs, BnaA9.HHO1 and BnaA1.HHO5 might play a core role in regulating rapeseed resistance to nutrient stresses. However, additional investigations are required to confirm the functional roles of these core genes. (Page 23, lines 694-697)
Comment 13: In one supplementary excel sheet, I suggest authors provide all the gene's names, original IDs, and protein sequences.
Response: Thank you very much for your kind suggestion. In view of this valuable suggestion, we have added this part to the Supplementary Materials of the revised materials as shown below.
Once again, special thanks for your valuable comment and kind suggestion.

Reviewer 2 Report
To,
The Editor,
IJMS, MDPI,
Manuscript ID: ijms-2004886
Subject: Submission of comments of the manuscript in “IJMS"
Dear Editor IJMS, MDPI,
Thank you very much for the invitation to consider a potential reviewer for the manuscript (ID: ijms-2004886). My comments responses are furnished below as per each reviewer’s comments.
In the reviewed manuscript, the authors identified 146 BnaGARP members in Brassica napus. The gene structure analysis revealed that each GARP gene in B. napus was found to be highly conserved. Conserved motif analysis showed that the BnaGARPs all contained the B-motif. Analysis of the Ka/Ks ratios indicated that the paralogs of the GARP family principally underwent purifying selection. Further, Cis-element analyses of the GARP genes identified 21 types of cis-elements in response to environmental stress and plant phytohormone. Furthermore, differential expression of BnaGARPs under low nitrate, ammonium toxicity, limited phosphate, deficient boron, salt stress condition, and cadmium toxicity conditions indicated their potential involvement in diverse nutrient stress responses, and several BnaNIGT1/HRS1/HHO transcription factors were found to be involved in N-starvation responses. This study increases the understanding of the evolution of the GARP gene family and provides valuable candidate genes for further study of the transcriptional regulation mechanism in response to nitrogen starvation in rapeseed. In general, the manuscript represents a very big piece of information in a logical presentation. The study is well-conducted and provided important results that might use to improve diverse nutrient stress in plants. Therefore, it might be conditionally accepted subject to major revision. Authors have to improve their manuscripts with many non-clear meanings, inaccuracies and inconsistencies, and the authors need to address the following issues before it can be accepted for publication.
1. I have read the entire manuscript and my initial comment is that manuscript is poorly written. I have significant concerns about the grammar and vocabulary of the manuscript; therefore, I recommend the authors to used an English proofreading service.
2. The structure of the abstract should be improved, as well as the lack of several aspects that should be included in this section. Most of the abstracts contain confusing and uninformative sentences. Please give more precise objectives here (such as in the Abstract). The abstract should highlight the most important results of the parameters and characteristics assayed.
3. Introduction grammatical issues appear to be most prevalent in the introduction, making for very confusing reading. Further, the introduction is short but has no clear thread.
4. General note: the figures in this section are quite low resolution and difficult to make out. Higher-resolution versions will be needed for publication, for example, in Figures 1, 2A, 2B, 2C, 2D, 3A, 3B, 4, 5, 6, 7, 8E, 8F, 9B, 9C, and Figure S1, S2.
5. Authors must examine the syntenic relationships with other model crops for instance, rice, wheat, Medicago truncatula, G. max etc
6. Why do authors carry out syntenic relationships with Arabidopsis to B. napus two time? They must reanalyze with B. napus vs Arabidopsis; B. napus vs rice; B. napus vs wheat; B. napus vs Medicago truncatula and B. napus vs G. max.
7. Why you selected this crop for your experiment? Please provide the detail of the used variety.
8. In Material and Methods:- indicate how many replicates assayed in each analysis/parameter. The number of samples or biological and technical replicates should be mentioned for each parameter in the methods.
9. Material methods most the citation is the webpage, some website is not working, hence, better to cite the original research paper.
10. Results must be explained clearly and in detail.
11. Why authors not verified the expression data with qRT-PCR?
12. The discussion should be interpreted with the results as well as discussed in relation to the present literature and authors must cite recently published research articles in the introduction and discussion on genome-wide identification in other mode crops, for instance, "https://www.ncbi.nlm.nih.gov/pmc/articles/PMC8880425/
https://www.frontiersin.org/articles/10.3389/fpls.2021.663118/full
https://pubmed.ncbi.nlm.nih.gov/33673010
https://www.ncbi.nlm.nih.gov/pmc/articles/PMC8395832
https://bmcplantbiol.biomedcentral.com/articles/10.1186/s12870-020-02576-0
https://www.frontiersin.org/articles/10.3389/fpls.2021.748146/ful
13. The conclusion section is very poorly written. It should be extensively improved.
14. References: shall have to correct the whole References according to the ”Instructions for the Authors”, e.g. the Journal name is in italics, the year must be bold and you shall have to use the abbreviated number of the Journals cited.
Author Response
Point-to-point response to Reviewer 2#
General comment: In the reviewed manuscript, the authors identified 146 BnaGARP members in Brassica napus. The gene structure analysis revealed that each GARP gene in B. napus was found to be highly conserved. Conserved motif analysis showed that the BnaGARPs all contained the B-motif. Analysis of the Ka/Ks ratios indicated that the paralogs of the GARP family principally underwent purifying selection. Further, Cis-element analyses of the GARP genes identified 21 types of cis-elements in response to environmental stress and plant phytohormone. Furthermore, differential expression of BnaGARPs under low nitrate, ammonium toxicity, limited phosphate, deficient boron, salt stress condition, and cadmium toxicity conditions indicated their potential involvement in diverse nutrient stress responses, and several BnaNIGT1/HRS1/HHO transcription factors were found to be involved in N-starvation responses. This study increases the understanding of the evolution of the GARP gene family and provides valuable candidate genes for further study of the transcriptional regulation mechanism in response to nitrogen starvation in rapeseed. In general, the manuscript represents a very big piece of information in a logical presentation. The study is well-conducted and provided important results that might use to improve diverse nutrient stress in plants. Therefore, it might be conditionally accepted subject to major revision. Authors have to improve their manuscripts with many non-clear meanings, inaccuracies and inconsistencies, and the authors need to address the following issues before it can be accepted for publication.
Response: We appreciate for your warm work and thanks very much for your positive and constructive comments on our manuscript. Your comments are considerably valuable and very helpful for revising and improving our paper, particularly on the experimental design for proceeding the project in the future. We have studied your comments carefully and have made corrections in red with tracked changes which we hope meet with your approval. The point-by-point responses to your comments are also listed as below.
Comment 1: I have read the entire manuscript and my initial comment is that manuscript is poorly written. I have significant concerns about the grammar and vocabulary of the manuscript; therefore, I recommend the authors to used an English proofreading service.
Response: Thank you very much for your kind suggestion. We have tried our best to polish the language in the revised manuscript. These changes will not influence the content and framework of this paper. And here we did not list the changes but marked in red in the revised manuscript.
Comment 2: The structure of the abstract should be improved, as well as the lack of several aspects that should be included in this section. Most of the abstracts contain confusing and uninformative sentences. Please give more precise objectives here (such as in the Abstract). The abstract should highlight the most important results of the parameters and characteristics assayed.
Response: Thank you very much for your kind suggestion. In view of this valuable suggestion, we have readjusted the structure and content of the Abstract of the revised manuscript, which is as follows:
Abstract: The GARP genes are a plant-specific transcription factors (TFs) and play key roles in regulating plant development and abiotic stress resistance. However, few systematic analyses of GARPs have been reported in allotetraploid rapeseed (Brassica napus L.) yet. In the present study, a total of 146 BnaGARP members were identified from the rapeseed genome based on the se-quence signature. The BnaGARP TFs were divided into five subfamilies: ARR, GLK, NIGT1/HRS1/HHO, KAN, and PHL subfamilies, and the members within the same subfamilies shared the similar exon-intron structures and conserved motif configuration. Analyses of the Ka/Ks ratios indicated that the GARP family principally underwent purifying selection. Several cis-acting regulatory elements, essential for plant growth and diverse biotic and abiotic stresses, were identified in the promoter regions of BnaGARPs. Further, 29 putative miRNAs were identified to be targeting BnaGARPs. Differential expression of BnaGARPs under low nitrate, ammonium toxicity, limited phosphate, deficient boron, salt stress, and cadmium toxicity conditions indicated their potential involvement in diverse nutrient stress responses. Notably, BnaA9.HHO1 and BnaA1.HHO5 were simultaneously transcriptionally responsive to these nutrient stresses in both hoots and roots, which indicated that BnaA9.HHO1 and BnaA1.HHO5 might play a core role in regulating rapeseed resistance to nutrient stresses. Therefore, this study would enrich our understanding of molecular characteristics of the rapeseed GARPs and will provide valuable candidate genes for further in-depth study of the GARP-mediated nutrient stress resistance in rapeseed.
Comment 3: Introduction grammatical issues appear to be most prevalent in the introduction, making for very confusing reading. Further, the introduction is short but has no clear thread.
Response: Thank you very much for your kind suggestion. We apologize for the confusion generated by the previous version of the manuscript and rephrased this part in our latest version.
Comment 4: General note: the figures in this section are quite low resolution and difficult to make out. Higher-resolution versions will be needed for publication, for example, in Figures 1, 2A, 2B, 2C, 2D, 3A, 3B, 4, 5, 6, 7, 8E, 8F, 9B, 9C, and Figure S1, S2.
Response: Thank you very much for your kind suggestion. According to your advice, we have improved the quality of the figures in the Figure zip file that we submitted.
Comment 5: Authors must examine the syntenic relationships with other model crops for instance, rice, wheat, Medicago truncatula, G. max etc
Response: Thank you very much for your kind suggestion. In view of this valuable suggestion, we have added this part to the Results of the revised manuscript, which is as follows:
…
To better understand the evolution of BnaGARP genes, the synteny of the GARP gene pairs between the genomes of B. napus and A. thaliana, G. max, and M. truncatula was constructed (Figure 5 and Table S4-S8). (Page 6, lines 197-199)
Comment 6: Why do authors carry out syntenic relationships with Arabidopsis to B. napus two time? They must reanalyze with B. napus vs Arabidopsis; B. napus vs rice; B. napus vs wheat; B. napus vs Medicago truncatula and B. napus vs G. max.
Response: Thank you very much for your kind suggestion. In view of this valuable suggestion, we have corrected this part to the Results of the revised manuscript, which is as follows:
…
As shown in Figure 5 and Figure S5, BnaGARP genes shared 172 syntenic gene pairs with G. max, 66 with M. truncatula, and 3 with T. aestivum (Table S5-S7). Additionally, 1 syntenic gene pairs were identified between rapeseed and rice, which constituted the fewest number of background collinear blocks (Table S8). Interestingly, 46 genes were found between B.napus and other plants (A. thaliana, M. truncatula, and G. max ) comparative synteny maps, and these collinear gene pairs were highly conserved within several syntenic blocks, such as BnaA1.APRR2, BnaA1.HHO5, BnaA1.KAN3, BnaA1.PHR1, and BnaA10.MYR1 on A1 chromosome and BnaA3.APRR2, BnaA3.ARR10, and BnaA3.ARR2 on A3 chromosome. (Page 6, lines 202-210)
Comment 7: Why you selected this crop for your experiment? Please provide the detail of the used variety.
Response: Thank you very much for your kind suggestion. We selected this crop because it is a major oilseed crop due to its economic value and oilseed production. However, its productivity has been reduced by many environmental adversities. To date, GARP genes have been identified to play a role in plant growth and response to stress. However, few systematic analyses of GARPs in B. napus have been available so far. Thus, We carried out this study on GARP genes in rapeseed. In addition, we provided the detail information about this species, which is as follows:
…
The B. napus seedlings (Darmor-bzh) were germinated in this experiment. “Darmor-bzh” is a French winter oilseed rape variety, whose reference genome sequence was first published in 2014 [109]. (Page 22, lines 639-641)
Comment 8: In Material and Methods: indicate how many replicates assayed in each analysis/parameter. The number of samples or biological and technical replicates should be mentioned for each parameter in the methods.
Response: Thank you very much for your kind suggestion. In the revised manuscript, we have added the number and replicates of samples in the Materials and Methods according to your suggestions, which is as follows:
…
The shoots and roots of fresh rapeseed seedlings above-mentioned were sampled separately and were immediately stored at 80 ℃. Each sample contained three independent biological replicates for the transcriptional analyses of BnaGARPs under di-verse nutrient stresses.
A total of 12 RNA samples from each treatment were subjected to an Illumina HiSeq X Ten platform (Illumina Inc., San Diego, CA, USA). (Page 22, lines 671-677)
Comment 9: Material methods most the citation is the webpage, some website is not working, hence, better to cite the original research paper.
Response: Thank you very much for your kind suggestion. Based on your suggestion, we have checked all the websites and added the original research paper of the webpage in our resubmitted manuscript, some of the changes are as follows:
…
In this study, the genomic, coding sequences, and protein sequences from A. thaliana and B. napus (Brana_ Dar_V5 genome) were downloaded from the Arabidopsis Information Resource (TAIR10, https://www.arabidopsis.org/, accessed on 01 October 2022) [90] (Page 20, lines 577-579)
Comment 10: Results must be explained clearly and in detail.
Response: Thank you very much for your kind suggestion. We have modified the structure and content in the Results of the revised manuscript.
Comment 11: Why authors not verified the expression data with qRT-PCR?
Response: Thank you very much for your kind suggestion.
As you have suggested, the expression profiling obtained from the RNA-seq data used in this study should be confirmed by the qRT-PCR assays. Indeed, the transcriptomic data have been demonstrated to be highly accurate and credible in many published papers of our group, some of which are as follows:
Zhou T, Yue CP, Liu Y, Zhang TY, Huang JY, Hua YP. Multiomics reveal pivotal roles of sodium translocation and compartmentation in regulating salinity resistance in allotetraploid rapeseed. J Exp Bot. 2021 Jul 28;72(15):5687-5708. doi: 10.1093/jxb/erab215.
Zhang ZH, Zhou T, Tang TJ, Song HX, Guan CY, Huang JY, Hua YP. A multiomics approach reveals the pivotal role of subcellular reallocation in determining rapeseed resistance to cadmium toxicity. J Exp Bot. 2019 Oct 15;70(19):5437-5455. doi: 10.1093/jxb/erz295.
Feng YN, Cui JQ, Zhou T, Liu Y, Yue CP, Huang JY, Hua YP. Comprehensive dissection into morpho-physiologic responses, ionomic homeostasis, and transcriptomic profiling reveals the systematic resistance of allotetraploid rapeseed to salinity. BMC Plant Biol. 2020 Nov 24;20(1):534. doi: 10.1186/s12870-020-02734-4.
Zhou T, Yue CP, Huang JY, Cui JQ, Liu Y, Wang WM, Tian C, Hua YP. Genome-wide identification of the amino acid permease genes and molecular characterization of their transcriptional responses to various nutrient stresses in allotetraploid rapeseed. BMC Plant Biol. 2020 Apr 8;20(1):151. doi: 10.1186/s12870-020-02367-7.
Zheng LW, Ma SJ, Zhou T, Yue CP, Hua YP, Huang JY. Genome-wide identification of Brassicaceae B-BOX genes and molecular characterization of their transcriptional responses to various nutrient stresses in allotetraploid rapeseed. BMC Plant Biol. 2021 Jun 24;21(1):288. doi: 10.1186/s12870-021-03043-0.
Cui JQ, Hua YP, Zhou T, Liu Y, Huang JY, Yue CP. Global Landscapes of the Na+/H+ Antiporter (NHX) Family Members Uncover their Potential Roles in Regulating the Rapeseed Resistance to Salt Stress. Int J Mol Sci. 2020 May 12;21(10):3429. doi: 10.3390/ijms21103429.
Wang Y, Hua YP, Zhou T, Huang JY, Yue CP. Genomic identification of nitrogen assimilation-related genes and transcriptional characterization of their responses to nitrogen in allotetraploid rapeseed. Mol Biol Rep. 2021 Aug;48(8):5977-5992. doi: 10.1007/s11033-021-06599-0.
Liu Y, Hua YP, Chen H, Zhou T, Yue CP, Huang JY. Genome-scale identification of plant defensin (PDF) family genes and molecular characterization of their responses to diverse nutrient stresses in allotetraploid rapeseed. PeerJ. 2021 Sep 13;9:e12007. doi: 10.7717/peerj.12007.
All of these papers have shown that there is a high consistency between the qRT-PCR assays and RNA-seq data. In addition, as severely affected by the Covid-19 in Zhengzhou (Henan province, China), currently, we could not perform the qRT-PCR assays due to the temporary closed-off management of our lab.
Comment 12: The discussion should be interpreted with the results as well as discussed in relation to the present literature and authors must cite recently published research articles in the introduction and discussion on genome-wide identification in other mode crops, for instance, "https://www.ncbi.nlm.nih.gov/pmc/articles/PMC8880425/
https://www.frontiersin.org/articles/10.3389/fpls.2021.663118/full
https://pubmed.ncbi.nlm.nih.gov/33673010
https://www.ncbi.nlm.nih.gov/pmc/articles/PMC8395832
https://bmcplantbiol.biomedcentral.com/articles/10.1186/s12870-020-02576-0
https://www.frontiersin.org/articles/10.3389/fpls.2021.748146/ful
Response: We sincerely appreciate the valuable comments. We have revised the content of the discussion in the revised manuscript. (Pages 17-20, lines 410-574)
Comment 13: The conclusion section is very poorly written. It should be extensively improved.
Response: Thank you very much for your kind suggestion. According to your suggestion, we have rewritten this part in the revised manuscript. (Page 23, lines 692-700)
Comment 14: References: shall have to correct the whole References according to the ”Instructions for the Authors”, e.g. the Journal name is in italics, the year must be bold and you shall have to use the abbreviated number of the Journals cited.
Response: Thank you very much for your kind suggestion. We are very sorry for this mistake. According to the “Instructions for the Authors”, we have corrected the whole references.
Once again, special thanks for your valuable comment and kind suggestion.

Round 2
Reviewer 1 Report
The authors have made a significant improvement and the revised version can be accepted for publication.
Reviewer 2 Report
Dear Editor,
Thank you for providing the opportunity to review the revised manuscript. The manuscript is improved considerably after revision according to the reviewer's comment. Now this study is a suitable contribution to the IJMS. I recommend the manuscript for publication.
Thank you
With best regards